

# Aerosol data assimilation in the chemical-transport model MOCAGE during the TRAQA/ChArMEx campaign: Aerosol optical depth

Bojan Sič[1,2], Laaziz El Amraoui[1], Andrea Piacentini[2], Virginie Marécal[1], Emanuele Emili[2], Daniel Cariolle[2], Michael Prather[3], and Jean-Luc Attié[4,1]

[1]CNRM-GAME, Météo-France – CNRS, UMR3589, Toulouse, France
[2]CECI, CERFACS – CNRS, UMR5318, Toulouse, France
[3]Department of Earth System Science, University of California, Irvine, USA
[4]Laboratoire d'Aérologie, University of Toulouse – CNRS, UMR5560, Toulouse, France

*Correspondence to:* B. Sič (sic@cerfacs.fr)

**Abstract.**

In this study, we describe the development of the aerosol optical depth (AOD) assimilation module in the chemistry-transport model (CTM) MOCAGE (*fr.* Modèle de Chimie Atmosphérique à Grande Echelle). Our goal is to assimilate the 2D column AOD data from the National Aeronautics and Space Administration (NASA) Moderate-resolution Imaging Spectroradiometer
(MODIS) instrument and to estimate improvements in a 3D CTM assimilation run compared to a direct model run. Our assimilation system uses 3D-FGAT (First Guess at Appropriate Time) as an assimilation method and the total 3D aerosol concentration as a control variable. In order to have an extensive validation data set, we set our experiment in the northern summer of 2012 when the pre-ChArMEx (CHemistry and AeRosol MEditerranean EXperiment) field campaign TRAQA (*fr.* TRAnsport à longue distance et Qualité de l'Air dans le bassin méditerranéen) took place in the western Mediterranean
basin. The assimilated model run is evaluated independently against a range of aerosol properties (2D and 3D) measured by in-situ instruments (the TRAQA size-resolved balloon and aircraft measurements), the satellite Spinning Enhanced Visible and InfraRed Imager (SEVIRI) instrument and ground-based instruments from the Aerosol Robotic Network (AERONET) network. The evaluation demonstrates that the AOD assimilation greatly improves aerosol representation in the model. For example, the comparison of the direct and the assimilated model run with AERONET data shows that the assimilation reduced
the bias in the AOD (from 0.050 to 0.006) and increased the correlation (from 0.74 to 0.88). When compared to the 3D concentration data obtained by the in-situ aircraft and balloon measurements, the assimilation consistently improves the model output. The best results as expected occur when the shape of the vertical profile is correctly simulated by the direct model. We also examine how the assimilation can influence the modelled aerosol vertical distribution. The results show that a 2D continuous AOD assimilation can improve the 3D vertical profile, as a result of differential horizontal transport of aerosols in
the model.



# 1 Introduction

In recent years, the role of aerosols in the climate system has been better determined (Boucher et al., 2013). As a consequence, efforts to accurately represent aerosols in models also increased (Textor et al., 2006; Lee et al., 2011). The development of aerosol modelling enabled us to better understand how aerosols affect atmospheric chemistry, air quality, climate, aviation,

visibility, radiative budget and clouds. Still, the complexity of the processes governing aerosol physics and chemistry has led to a large diversity of parameterizations which produce large differences in the aerosol model results (Mahowald et al., 2003; Kinne et al., 2006; Textor et al., 2006; Shindell et al., 2013).

At the same time, the number and quality of aerosol observations has also increased through advances in sensing technology and techniques. The last decade of aerosol research has brought on more accurate measurements of more specific aerosol

characteristics observed from local to global scales and over long periods of time (De Leeuw et al., 2011). In essence, we are in a golden age of aerosol data, but still looking for approaches to merge all the disparate measurements and synthesize that knowledge (Boucher et al., 2013).

Observations are crucial in identifying aerosol properties and processes, which in turn help building more accurate models. Also, with data assimilation techniques, we are able to directly integrate observations in models in order to improve modelled

fields. Up to now, several research groups made efforts to assimilate aerosols in the models. These efforts are mainly focused on assimilating satellite data, usually aerosol optical depth (AOD), since satellites provide continuous aerosol observations on the global scale and yield a large number of individual observations which is desirable for assimilation systems. Many studies used variational data assimilation techniques, with a 3D-VAR approach Zhang et al. (2008); Niu et al. (2008); Liu et al. (2011) or a 4D-Var approach Benedetti et al. (2009). Sequential assimilation approaches are also documented, by using, for example,

optimal interpolation (Tombette et al., 2009) or the Ensemble Kalman filter (Schutgens et al., 2010; Pagowski and Grell, 2012; Dai et al., 2014).

In this study, we describe the development of the AOD assimilation module in the chemistry-transport model (CTM) MOCAGE (e.g. Josse et al., 2004; Sič et al., 2015, *fr*: Modèle de Chimie Atmosphérique à Grande Echelle;). Our goal is to assimilate the regular, daily global mapping of the 2D column AOD by the National Aeronautics and Space Administration

(NASA) Terra and Aqua satellites (Remer et al., 2005) with the 3D CTM modelling of the major tropospheric aerosols. We use the variational 3D-FGAT (First Guess at Appropriate Time) method (Andersson et al., 1998) which is implemented in the CTM using the PALM coupler (Buis et al., 2006; Massart et al., 2007). The assimilated fields can then be evaluated independently against a large range of aerosol properties (2D and 3D) measured by other satellites, in-situ and ground-based instruments. We will estimate improvements in the CTM modelling of all aerosol types brought by the AOD assimilation compared to a

direct (unassimilated) model run. One focus will be on the potential to improve air quality forecasting. Another will be how the continuous multicycle AOD assimilation can influence the modelled aerosol vertical distribution. To obtain an extensive validation data set we choose the 2012 summer field campaign of TRAQA (*fr*: TRAnsport à longue distance et Qualité de l'Air dans le bassin méditerranéen) and center our modeling on the western Mediterranean basin. The TRAQA campaign was a pre-ChArMEx (CHemistry and AeRosol MEditerranean EXperiment) experiment with an objective to characterize the air



quality in the western Mediterranean basin (Attié et al., in preparation). The validation data includes not only the TRAQA measurements (in-situ aircraft and balloon measurements of size-resolved and speciated aerosols), but also different remote-sensed AOD observations from the ground (Aerosol Robotic Network [AERONET], Holben et al. (1998)) and satellite (Spinning Enhanced Visible and InfraRed Imager [SEVIRI], Thieuleux et al. (2005)).

This paper is organized as follows. In Sect. 2 we describe the MOCAGE CTM and its aerosol module; in Sect. 3 the data assimilation system; and in Sect. 4 and Sect. 5, respectively, the assimilated and the independent observational data. Results from the assimilation model and its critical evaluation against the direct forecasts of the CTM with independent data are presented in Sect. 6. In Sect. 7 we discuss the overall performance of our assimilation system; and in Sect. 8 our recommendations for future work.

## 10   2    Model description

MOCAGE is a global CTM developed in Météo-France. It serves as an operational air quality model and simulates gases (Josse et al., 2004; Dufour et al., 2005) and primary (directly emitted) aerosols (Martet et al., 2009; Sič et al., 2015). It transports atmospheric species by a semi-lagrangian advection scheme (Williamson and Rasch, 1989). Turbulent diffusion is implemented following Louis (1979); and convection, following Bechtold et al. (2001). The dynamics within the CTM

are forced by meteorological analysis fields (pressure, winds, temperature, specific humidity) from ARPEGE, the operational numerical weather prediction model of Météo-France. MOCAGE has 47 vertical hybrid sigma-pressure levels from the surface up to 5 hPa. The vertical resolution varies with altitude, with a resolution of 40 m in the planetary boundary layer, about 400 m in the free troposphere and about 700–800 m in the upper troposphere and lower stratosphere. The version of the model used in this study is thoroughly described in Sič et al. (2015) and has been evaluated with a range of different remote-sensed and

in-situ measurements for cases and regions relevant to this study.

The model can include nested domains over smaller regions. In this study, the model is run in a two-domain configuration with a global grid of $2° × 2°$ and a smaller nested domain (MEDI02) with a grid of $0.2° × 0.2°$ over the Mediterranean basin and the Sahara desert. The MEDI02 domain, where we assimilate AOD data, has boundaries $20°$ W $- 40°$ E, $16°$ N $- 52°$ N. The lateral boundary conditions for aerosols are provided by the global domain.

Aerosols in MOCAGE consist of externally mixed primary aerosol species. Implemented species for this study are: desert dust, sea salt, black carbon (BC) and organic carbon (OC). The particle size distribution for each type is divided into 6 size bins, characterized by the particle average diameter and mass. Each aerosol bin is then treated as a passive tracer: aerosols are emitted, transported and removed from the atmosphere, however there are no transformations or chemical reactions between aerosol types, between size bins or with gases. The aerosol dry deposition scheme is described in detail in Nho-Kim et al.

(2004). The sedimentation is implemented as described in Seinfeld and Pandis (1998). For the wet deposition, the model uses Giorgi and Chameides (1986) for the implementation of the in-cloud scavenging, and Slinn (1977) for the rain and snowfall below-cloud scavenging.



The emission inventories for BC and OC are prepared as follows. The anthropogenic component comes from Lamarque et al. (2010) for both domains (Global and MEDI02). This inventory is defined monthly, and harmonized for the year 2000 (Lamarque et al., 2010). Biomass burning (BB) sources of BC and OC aerosols are introduced into the model with a daily frequency from the Global Fire Assimilation System (GFAS) version 1.1 (Kaiser et al., 2012). The GFAS assimilates the fire

radiative power observed by MODIS, corrects the cloud cover gaps, filters anthropogenic and volcanic activities, and finally calculates daily biomass burning aerosol emissions for BC and OC. We use daily BB emissions for better synoptic forecasts, which is not possible with the monthly mean emissions of Lamarque et al. (2010).

Sea salt particles are emitted using the semi-empirical source function from Jaeglé et al. (2011) which includes a particle size, wind speed, and sea surface water temperature dependence. Desert dust aerosols are emitted by a dynamical online

scheme which depends on the wind intensity and surface characteristics. The scheme is based on Marticorena et al. (1997). It covers Africa, Arabia and the Middle East [13–36° N, 17° W–77° E], where input soil properties and aerodynamical surface parameters have a resolution of $1° \times 1°$ (Marticorena et al., 1997), and north-eastern Asia [35.5–47° N, 73–125° E] with the input parameter resolution of $0.25° \times 0.25°$ (Laurent et al., 2006).

## 3 Description of data assimilation system

The data assimilation system used in this study is MOCAGE-Valentina, developed jointly by Météo-France and CERFACS (Centre Européen de Recherche et de Formation Avancée en Calcul Scientifique). The used assimilation algorithm is 3D-FGAT (3-Dimensional First Guess at Appropriate Time; Fisher and Andersson, 2001; Massart et al., 2010), which is a compromise between the 3D-Var and 4D-Var methods. Observations are taken at their exact times to the nearest minute, i.e. every measurement is compared with the background at the time of measurement, as in 4D-Var. The optimal analysis is estimated only

for a specified moment in the assimilation cycle, as in 3D-Var, and not for the whole trajectory, as in 4D-Var. Thus, during the assimilation, we do not need the linearized operator of the model evolution and its adjoint, as in 4D-Var. In 3D-FGAT the information given by observations is not propagated in time as in 4D-VAR (Courtier et al., 1994).

The goal of the assimilation process is to minimize the cost function, whose incremental form in 3D-FGAT is:

$$J(\delta x) = J_b(\delta x) + J_o(\delta x) = \frac{1}{2} \delta x^{\mathrm{T}} \mathbf{B}^{-1} \delta x + \frac{1}{2} \sum_{i=0}^{N} (d_i - \mathbf{H}_i \delta x)^{\mathrm{T}} \mathbf{R}_i^{-1} (d_i - \mathbf{H}_i \delta x), \tag{1}$$

where $J_b$ is a part of the cost function related to the background; $J_o$ is a part of the cost function related to the observations; $\delta x = x - x^b$ is the misfit between the background $x^b$ and the state of the system $x$; $d_i = y_i - H_i x^b(t_i)$ is the innovation and represents the distance of the observation $y_i$ from the background $x^b$ at time $t_i$; $H_i$ is the non-linear observation operator; $\mathbf{H}$ is its linearized version (tangent-linear); $\mathbf{B}$ is the background error covariance matrix; and $\mathbf{R}_i$ is the observation error covariance matrix at time $t_i$. The matrices $\mathbf{B}$ and $\mathbf{R}_i$ influence the weighting of the terms $J_b$ and $J_o$.

To find the optimal solution we minimize the cost function $J$ by computing its gradient:

$$\nabla J(\delta x) = \mathbf{B}^{-1} \delta x + \sum_{i=0}^{N} \mathbf{H}_i^{\mathrm{T}} \mathbf{R}_i^{-1} (d_i - \mathbf{H}_i \delta x). \tag{2}$$





After estimating the analysis increment $\delta x^a$, we add it to the aerosol abundance at the beginning of the cycle. The model is then run over a cycle length (1 hour) to obtain the analysed trajectory. Its endpoint is used as the initial background field for the next cycle.

## 3.1 Preconditioning

MOCAGE-Valentina uses the incremental form of 3D-FGAT (Eq. 1). In order to minimize the cost function more efficiently and to improve convergence, the increment $\delta x$ is preconditioned by:

$$v = \mathbf{B}^{-\frac{1}{2}} \delta x. \tag{3}$$

In this way the cost function becomes

$$J(x) = \frac{1}{2} v^{\mathrm{T}} v + \frac{1}{2} \sum_{i=1}^{N} (d_i - \mathbf{H}_i \mathbf{B}^{\frac{1}{2}} v)^{\mathrm{T}} \mathbf{R}_i^{-1} (d_i - \mathbf{H}_i \mathbf{B}^{\frac{1}{2}} v), \tag{4}$$

and its gradient

$$\nabla J(\delta x) = v + (\mathbf{B}^{\frac{1}{2}})^T \sum_{i=1}^{N} \mathbf{H}_i^T \mathbf{R}_i^{-1} (d_i - \mathbf{H}_i \mathbf{B}^{\frac{1}{2}} v). \tag{5}$$

In this formulation, there is no need for an explicit specification of the inverse matrix $\mathbf{B}^{-1}$, and the preconditioning reduces the number of iterations Courtier et al. (1994). In MOCAGE-Valentina, the cost function is minimized using the limited-memory BFGS (Broyden–Fletcher–Goldfarb–Shanno) method (Liu and Nocedal, 1989).

The minimization of the cost function with the preconditioned form gives an increment of the analysis in the space of the variable $v$, which after the minimization is converted back to model space:

$$\delta x = \mathbf{B}^{\frac{1}{2}} v. \tag{6}$$

More details on the assimilation algorithm are provided by Pannekoucke and Massart (2008) and Massart et al. (2012).

## 3.2 The control variable

For aerosols, the modelled prognostic variables (i.e., the 3D concentration of aerosols of different composition) and the observations (i.e., the column optical depth summed over all aerosols at visible wavelengths such as $550\,\mathrm{nm}$ – Aerosol Optical Depth or AOD) are usually not the same physical quantity. In the case of AOD observations, to define which variable will be minimized, different choices are possible. A straightforward choice is to use a prognostic variable as a control variable, as implemented by Liu et al. (2011). Following this approach in our system, the control variable would correspond to a 4D variable containing the 3D fields of all 24 aerosol bins. The matrix $\mathbf{B}$ would have to include the variances and covariances of all bins separately. This could be difficult to define, but the analysis would be partitioned automatically into all bins by the system.

Benedetti et al. (2009) made the choice to use the 3D total aerosol concentration as the control variable. This makes the control variable smaller, corresponding to a 3D variable, where all bins are merged into a single one. Considering the characteristics of our system, we decided to follow the same approach as in Benedetti et al. (2009). Compared to the Liu et al.





(2011) approach, the problem of minimization of the cost function is better determined: the 2D AOD observations constrain one 3D variable as the unknown, compared to 24 3D variables as unknowns in the Liu et al. (2011) approach. The matrix **B** does not have to be defined for all bins separately, nor it contains the inter-bin covariances. Nevertheless, in order to linearize the observation operator, it is necessary to decide how the analysis increment $\delta x^a$ will influence each bin. The increment could

be weighted by different quantities, like number or mass concentration, or extinction coefficient. The real contribution of different aerosol types in the increment is unknown and we can rely only on the model information. In this way, all repartition weights based on the model stay biased in a similar way compared to the real repartition weights, regardless of the choice of the repartition. Considering all possible choices and the characteristics of our system, we decided to keep constant the relative mass contributions. After the analysis increment is calculated, it is repartitioned to the different bins in the model according to

their background fractions of the total aerosol mass.

### 3.3 The observation operator

The assimilation of AOD in MOCAGE-Valentina requires the development of an observation operator ($H$) which transforms the control variable from the model space, i.e. the total 3D concentration, into the observation space, i.e. AOD. The AOD ($\tau$) is calculated by taking into account bin number concentrations ($n_{bin}$) at a certain model level and the optical properties of

individual species calculated by the Mie code:

$$Hx = \sum_{bin} \sum_{lev} C_{ext}(D_p, \tilde{n}, \lambda) \Delta z_{lev} n_{bin} = \tau. \tag{7}$$

where $\Delta z_{lev}$ is the vertical thickness of the model level $lev$ [m], and $C_{ext}$ is the extinction cross-section [m$^{-1}$]. To calculate $C_{ext}$ in the model we use Wiscombe's Mie code scheme for spherical homogeneous particles (Wiscombe, 1979, revised 1996, 1980) and aerosol refractive indices from the Global Aerosol Data Set (GADS, Köpke et al., 1997) and Kirchstetter et al.

(2004) and also by taking into account the hygroscopicity of sea salt aerosols.

The tangent-linear operator ($\mathbf{H} = \frac{\partial H}{\partial x_i}$) is a linearized version of the non-linear observation operator $H$ around the system state $x$ and it consists of partial derivatives of $H$ with respect to all input variables. It gives a first-order approximation ($\delta \tau$) of the difference between the unperturbed ($Hx$) and the perturbed results ($H(x + \Delta x)$) of the non-linear operator $H$.

The tangent linear operator can be derived explicitly by the finite-difference method, but it is a computationally expensive

method. Instead, the tangent-linear operator can be considered as a sequence of linearized sub-operators of the non-linear observation operator and built piece by piece by differentiating separately each line of code or loop.

This approach is convenient by allowing to test the parts of the code separately. To provide a linearized trajectory around the model state $x$, the tangent linear operator has to satisfy:

$$\lim_{\delta x \to 0} \frac{H(x + \delta x) - Hx}{\mathbf{H}\delta x} = 1. \tag{8}$$

As long as the perturbation $\delta x$ is small enough that it stays close to the model state $x$, the test will give a value close to 1.

The adjoint operator ($\mathbf{H}^{\mathrm{T}}$) is the transpose of the tangent-linear operator and it satisfies:

$$\langle \mathbf{H}x, y \rangle = \langle x, \mathbf{H}^{\mathrm{T}}y \rangle, \tag{9}$$





where $\langle x, y \rangle$ represents the inner product of $x$ and $y$. Analogously as for the tangent linear operator, the adjoint operator can be considered as a sequence of operators. Each discrete operation in the tangent linear operator has a corresponding operation in the adjoint operator, but the order of execution is reversed.

### 3.4 The error covariance matrices

The background error covariance matrix is a key component of the data assimilation system. It defines the model errors and the spatial structure of the analysis. The background error covariance matrix $\mathbf{B}$ is a matrix of size $j \times j$, where $j$ is the size of the control variable. It can be represented as:

$$\mathbf{B} = \mathbf{\Xi} \mathbf{C} \mathbf{\Xi}^{T}, \tag{10}$$

where $\mathbf{\Xi}$ is the diagonal matrix of the square root of the variances, and $\mathbf{C}$ is the positive definite symmetric matrix of correla-
tions. In the case of the preconditioned cost function, the matrix is formulated as:

$$\mathbf{B}^{\frac{1}{2}} = \mathbf{\Xi} \mathbf{C}^{\frac{1}{2}}, \tag{11}$$

where $\mathbf{C}^{\frac{1}{2}}$ is the square root of the matrix $\mathbf{C}$. Because not enough information is available to explicitly estimate all correlation members, nor is there enough memory to store them, the matrix $\mathbf{B}$ is modelled as an operator. To estimate the product of the matrix $\mathbf{B}$ and a vector, MOCAGE-Valentina uses the integration of a generalized diffusion-type equation in a reduced space
Weaver and Courtier (2001).

The observation error covariance defines the observation and representativeness errors. These errors are considered to be non-correlated, which means that all non-diagonal members (covariances) in the matrix $\mathbf{R}$ are zero. The matrix $\mathbf{R}$ is reduced to its diagonal with the variances of measurements

$$\mathbf{R} = \mathbf{D}_y = \mathrm{diag}(\sigma_{obs}^2). \tag{12}$$

The background and observation error variances, located along the diagonal of $\mathbf{B}$ and $\mathbf{R}$, influence the weight of the model and observations in the cost function. In this study, we specified them as a percentage of the control variable for the first guess field and a percentage of measured AOD for the observations. We used the $\chi^2$ diagnostics to estimate optimal values for errors of these datasets (Ménard et al., 2000; Talagrand, 2003). The $\chi^2$ test is *a posteriori* diagnostic which defines properly specified errors if, for each assimilation window, it is true that:

$$E\left(\frac{2J_{min}}{p}\right) \sim 1, \tag{13}$$

where $E$ is the expectation (statistical average), $J_{min}$ is the value of the cost function at the minimum and $p$ is the number of observations. For this test, it is necessary to run the assimilation system for a prolonged period of time and, in the case of sufficient number of observations, the matrix $\mathbf{B}$ will not depend any more on its initial value. Because this method is computationally expensive, a very rigorous optimization of the errors is difficult to do. Therefore, we carried out several test





runs to determine optimal parameters for the matrices *a posteriori*. As the optimal parameters, we estimated that the percentage for the errors of the model should be twice as large as for the observations ($24\%$ and $12\%$ respectively).

Missing secondary aerosols in MOCAGE are considered as a possible underestimation of AOD in the model, and this is taken into account in the covariance matrices error definitions. Therefore, the possible smaller AOD values of the model are

compensated by a higher percentage for the error of the model. Also, it is considered that it is better to have overestimated errors in **B**, than vice-versa (Talagrand, 2003).

Covariances of the background error matrix, which influence the spread of the analysis to neighbouring gridboxes, are specified with constant correlation lengths in the horizontal and the vertical. The constant and homogeneous correlation lengths are modelled using a Gaussian function (Pannekoucke and Massart, 2008) in terms of geographic degrees for the horizontal

lengths, and in terms of number of model levels for the vertical lengths (Massart et al., 2009). The implemented horizontal correlation length is $0.4°$. For the vertical correlation, with the column integrated observations there is no explicit need for vertical correlation elements. But, in the matrix **B** preconditioned cost function (Eq. 13) it is advantageous to not have null vertical correlation lengths. Therefore, we apply a vertical correlation of one model level. The type of the correlation field influences the method by which the generalized diffusion-type equation is solved. In the case of constant correlation lengths in

the limited-area domain, the equation is solved by the finite-difference method.

## 4   Assimilated observations

The MODIS (Moderate-resolution Imaging Spectroradiometer) instruments observe atmospheric aerosols onboard Terra (since 2000) and Aqua (since 2002) from complementary sun-synchronous orbits. The Terra overpass time is around 10:30 local solar time at the Equator in its descending (daytime) node, and the Aqua overpass time is around 13:30 local solar time at the

equator in the ascending node. We use MODIS Aerosol Optical Depth Collection 5 retrievals at $550\,\mathrm{nm}$ from Terra and Aqua: the ocean product retrieved with the "best solution" and the reflectance corrected land product. Their predicted uncertainties are $\Delta\tau = \pm(0.03 + 0.05\tau)$ over oceans and $\Delta\tau = \pm(0.05 + 0.15\tau)$ over land (Remer et al., 2005). Over bright desert areas, we use the "Deep blue" MODIS product (Hsu et al., 2006). For the assimilation, we only considered the best quality data, with the highest possible quality flag.

MODIS L2 resolution of $10\,\mathrm{km} \times 10\,\mathrm{km}$ is approximately two times finer than the model resolution of $0.2° \times 0.2°$ over the control MEDI02 domain in which the assimilation is performed. We have no way of treating two separate AOD values within a model grid-cell at about the same time and so just average all observations in each grid-cell that occur on the same swath (making so-called *super-observations*, Daley, 1993). Modelled and observed AOD are then on the same 2D grid and the maximal number of observations per one hour slot over the whole domain is reduced in this way from $\approx 80000$ to $\approx 15000$.

MODIS data from the Terra and Aqua platforms are separated in time, except at high latitudes (not used here) and all AODs are binned in the MEDI02 grid at 5 min intervals.



## 5 Independent observations for evaluation

### 5.1 SEVIRI

SEVIRI (Spinning Enhanced Visible and InfraRed Imager) geostationary observations over oceans are retrieved at 550 nm by Thieuleux et al. (2005). The ICARE data center operationally implemented this algorithm and makes AOD data available on its website (www.icare.univ-lille1.fr). This product was evaluated against other satellite products and AERONET measurements by Thieuleux et al. (2005) and Breon et al. (2011). The instrument makes an image of the whole Earth disk every 15 minutes as seen from the equatorial geostationary orbit and at longitude 0°. We sample the SEVIRI AOD every hour and only use data over water, since the retrieval over dark surfaces is usually more accurate. The nadir horizontal resolution is 3 m, while over Europe it is ≈ 5 m and we average the SEVIRI AOD data that fall within the same modelled gridbox.

### 5.2 AERONET

AERONET (AErosol RObotic NETwork) measures ground-based AOD from hundreds of automated stations in the world with an accuracy of $\pm 0.01$ for a range of wavelengths (Holben et al., 1998). We use all available L1.5 data from different stations and interpolate it in logarithmic space to 550 nm (to harmonize wavelengths between different stations and with the model) by using available neighbouring wavelengths: 440 nm, 500 nm, 675 nm, 870 nm.

### 5.3 In-situ observations from the TRAQA campaign

TRAQA (*fr.* TRAnsport à longue distance et Qualité de l'Air dans le bassin méditerranéen) was a scientific project including a measurement campaign intended as a pre-ChArMEx (CHemistry and AeRosol MEditerranean EXperiment) experiment (http://charmex.lsce.ipsl.fr). It took place over the north-western Mediterranean basin during the northern summer 2012. The main objectives of TRAQA were studies of transport, ageing and mixing of the polluted air masses in and around the Mediterranean basin and their impact on air quality. From 26 June to 11 July seven intensive observation periods were conducted with the ATR-42 aircraft of Météo-France, atmospheric balloons (sounding and drifting) and ground instruments measuring trace gases and aerosols. During the campaign, a desert dust outbreak from Africa transported aerosols to the Mediterranean basin. It was well observed around 29 June with several different instruments (Attié et al., in preparation).

#### 5.3.1 PCASP

In our study, we use the data measured by the passive cavity aerosol spectrometer probe (PCASP) which was onboard the ATR-42 aircraft. The PCASP measures the aerosol concentration and the aerosol size distribution with its 30 channels (Strapp et al., 1992). PCASP measures particles with diameters from from 0.1 μm to 3 μm, with channel ranges and calibration methods reported by Cai et al. (2013). The high-frequency instrument data is averaged into 1 minute intervals for model comparison with a horizontal resolution of about 8 km at cruise speed.



### 5.3.2 LOAC

Finally, we also use data from LOAC (Light Optical Particle Counter) instruments (Renard et al., 2015) collected during TRAQA. LOAC is a light aerosol optical counter measuring aerosol number concentration in 20 size classes within a diameter range from $2\,\mu m$ to $100\,\mu m$ in the version of the instrument used in the TRAQA campaign. LOAC uses the technique of measuring aerosol at two scattering angles (Lurton et al., 2014; Renard et al., 2015). During TRAQA, LOAC was mounted on meteorological sounding balloons, where its vertical resolution depends on its measurement frequency. The processed data that we use in our analysis has a vertical resolution in the troposphere of about $0.3\,km$ to $0.4\,km$ which is similar to the model resolution.

## 6 Results

We run two MOCAGE configurations, one with and one without assimilation. The simulation without assimilation is referred as the *direct model run* and the simulation with assimilation as the *assimilation model run*. The simulated period for which we evaluate the model performance is for 19 days from 25.06.2012 until 13.07.2012. The model is run after a spin-up period of 45 days. The assimilation cycles in the experiment have a length of one hour. The cost function is minimized until the convergence criterion is reached, or when the maximum number of iterations of 200 is reached. The analysis increment is added at the beginning of each assimilation cycle.

### 6.1 Performance of the assimilation

In Fig. 1 we evaluate the impact of the data assimilation on the modelled fields (directly forecasted and assimilated fields for each 1h cycle) by comparing each of them with the observations which we assimilated, and this should be considered as a "sanity" check of the system. The figure represents the performance of the assimilation system and its ability to move the modelled field closer to the observed values. The assimilated model can more readily lower the overestimated values than to elevate the underestimated values. The statistics of the scatterplots against assimilated observations show increased correlation (from $0.54$ for the modelled fields, to $0.82$ for the assimilated fields), lowered root mean square error (from $0.27$ to $0.18$) and lowered standard deviation (from $0.27$ to $0.17$) demonstrating what we expect from assimilation.

### 6.2 Comparison with SEVIRI

The period of the TRAQA campaign is marked by two desert dust events coming from Africa. In the western Mediterranean basin, where the campaign took place, this produces elevated concentrations of desert dust aerosols from the Sahara desert. Figure 2 shows the desert dust event in the Mediterranean basin on a particular day (29 June) seen by the direct model run, the assimilation model run and the SEVIRI instrument, and it illustrates the impact of the assimilation on the modelled field. We look at a longer time period in Fig. 3 which represents the timeseries of the direct and assimilation model runs and the independent SEVIRI observations over the western Mediterranean basin. Table 1 shows the statistics corresponding to this



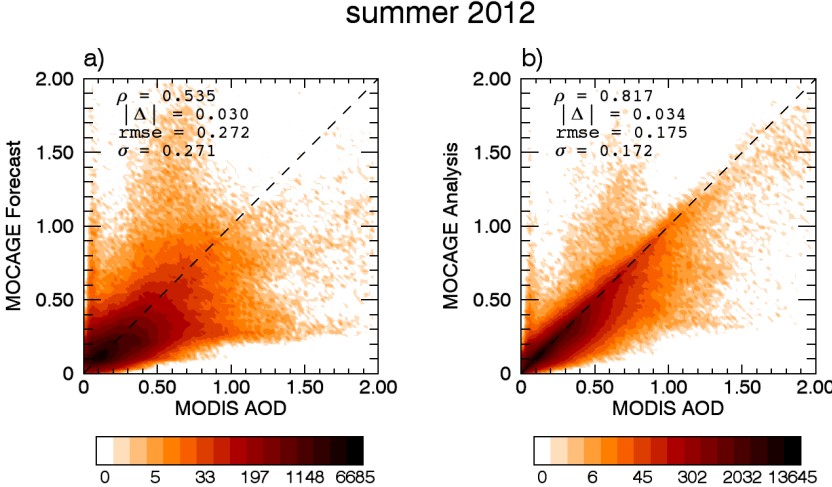

**Figure 1.** Scatterplots of aerosol optical depths (where colours represent the number of points) of assimilated MODIS observations and: the forecast **(a)**, and analysis **(b)** for each one hour assimilation window (both runs start from the same assimilated conditions of one hour before). In each panel, correlation ($\rho$), absolute bias ($\Delta$), root mean square error (RMSE) and standard deviation ($\sigma$) are noted. The assimilated data correspond to the period of the TRAQA campaign from 25.06.2012 until 13.07.2012., and covers the MEDI02 domain.

**Table 1.** Correlation ($\rho$), absolute bias ($\Delta$) and root mean square error (RMSE) between SEVIRI observations and MOCAGE direct/assimilation model run for the western Mediterranean during the TRAQA campaign between 25.06.2012 and 13.07.2012. The mean number of SEVIRI observations per hour is also given. The statistics correspond to Fig. 3 with the observations localized in the region 0–10° E, 35–45° N (marked in Fig. 5 by the grey box).

|  | $\overline{N_{obs}}$ [h$^{-1}$] | MOCAGE direct | | | MOCAGE assimilation | | |
|---|---|---|---|---|---|---|---|
|  |  | $\rho$ | $\Delta$ | RMSE | $\rho$ | $\Delta$ | RMSE |
| SEVIRI | 20875 | 0.83 | 0.14 | 0.17 | 0.96 | 0.08 | 0.09 |

figure. The two desert dust events in the figure are highlighted with high values of AOD ($> 0.25$). The first, stronger dust outbreak increases AOD values during 4 to 5 days over the region (from 27 June to 1 July). Its extent is well simulated in the model direct run, but its intensity is underestimated (Fig. 2). The assimilation produced fields which are closer to the SEVIRI observations. The second desert dust event occurs at the end of the TRAQA period (from 8 to 11 July). It is weaker than the first one and localised only in the part of the western Mediterranean closer to the coast of Africa. The AOD values of the second dust event, and also of the period between the two events, are underestimated in the direct model run, although not as strongly as during the first event. Data assimilation reduces the difference between the model and the observations, improving all statistical




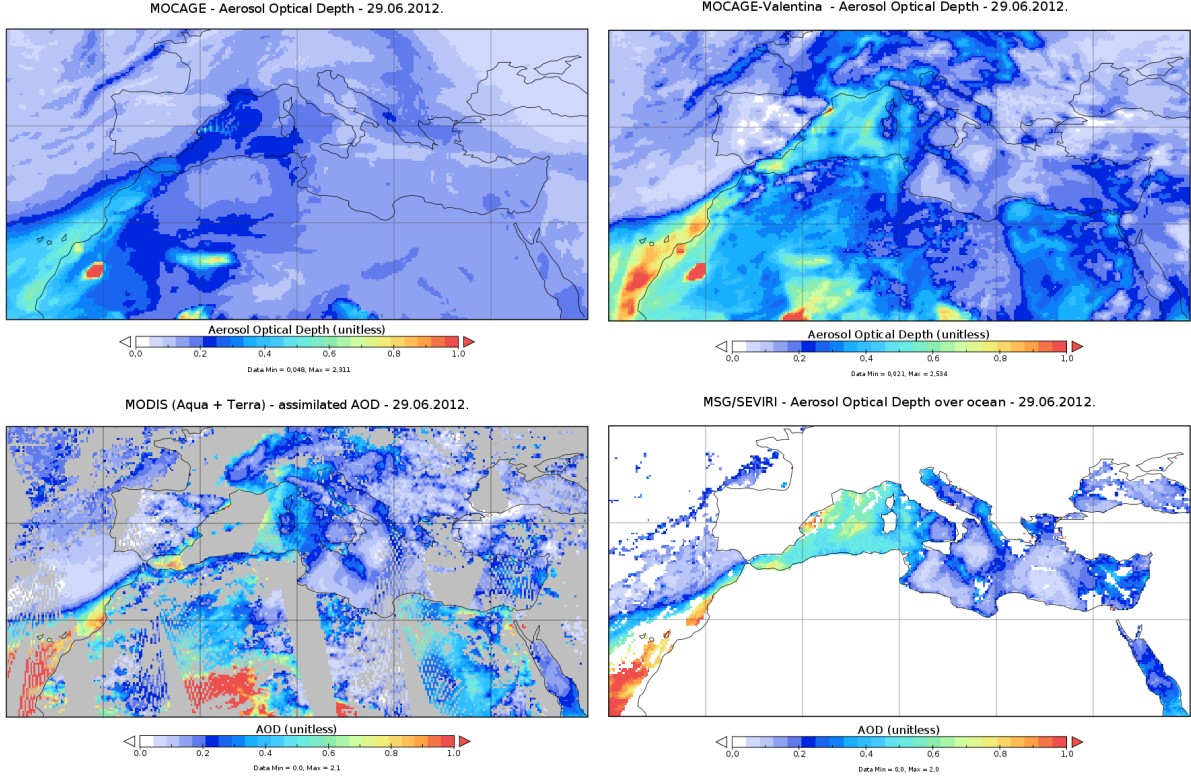

**Figure 2.** The aerosol optical depth over Europe on 29.06.2012 at 12h UT, (**top left**) simulated in MOCAGE by the model direct run; (**top right**) simulated in MOCAGE by the MODIS assimilation model run; (**botom left**) observed by MODIS (Aqua + Terra) and used for assimilation in MOCAGE (shown observations are collected during the whole day, and not only at 12h); and (**bottom right**) observed by SEVIRI that serves as an independent dataset. The colours from white to red represent AOD from low to high values.

parameters (Table 1). For example, the correlation improves from 0.83 for the direct model to 0.96 for the assimilated model run. MODIS overpasses each point twice during the daytime (approximately at 10.30 and 13.30 in local solar time), and this provides sufficient information to even improve the hourly AOD variation in the assimilated field during different dates (for example, 9-11 July; Fig. 3).

5    Figure 4 compares AOD from the direct and the assimilation model runs with SEVIRI observations over the whole control domain, and not only over the western Mediterranean. The majority of the observed points correspond to small AOD values where the direct model and the observations agree well. For larger observed values, the scatterplots confirm that the direct model run underestimated the AOD field, largely because the desert dust outbreaks were underestimated in the model. As expected, the assimilation reduced this disagreement, displaying also better statistics compared to the direct model run (e.g.

10   the correlation improved from 0.69 to 0.87).





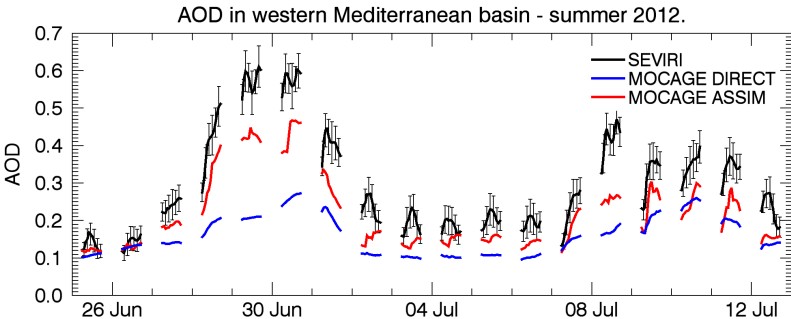

**Figure 3.** Hourly timeseries of aerosol optical depth at 550 nm of SEVIRI data, the direct model and the assimilation model run over the western Mediterranean (0–10° E, 35–45° N) for the period of the TRAQA campaign from 25.06.2012 until 13.07.2012. The considered region is also marked in Fig. 5 by the grey box. Correlation, bias and root mean square error for both the direct model and the assimilation model run as compared to SEVIRI data are given in Table 1.

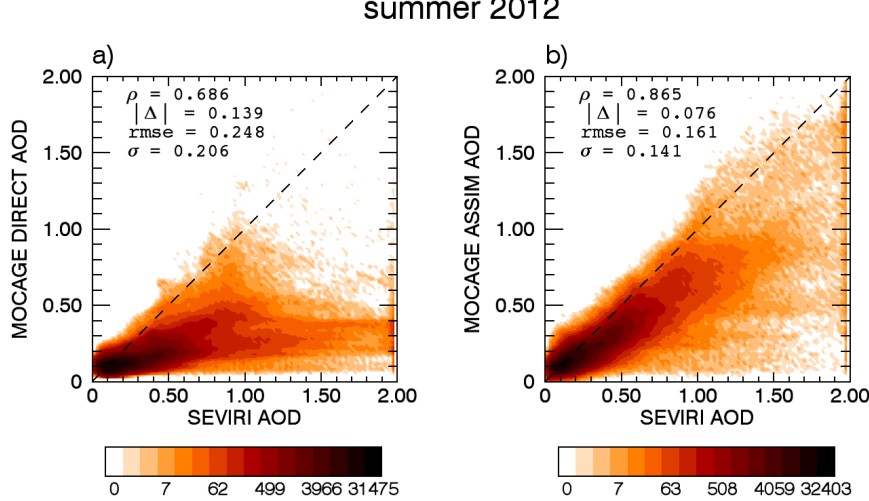

**Figure 4.** Scatterplots of aerosol optical depths (where colours represent the number of points) from the independent observation dataset (SEVIRI) and the simulations: the direct model run **(a)** and the assimilation model run **(b)**. In each panel, correlation ($\rho$), absolute bias ($\Delta$), root mean square error (RMSE) and standard deviation ($\sigma$) are noted. The included data correspond to the period of the TRAQA campaign from 25.06.2012 until 13.07.2012., and covers the whole MEDI02 domain.

## 6.3 Comparison with AERONET

We compare the model direct run and assimilation model run with the AOD data from AERONET stations. In total, we consider measurements from 35 AERONET stations which are all in or around the Mediterranean basin. Their locations are presented



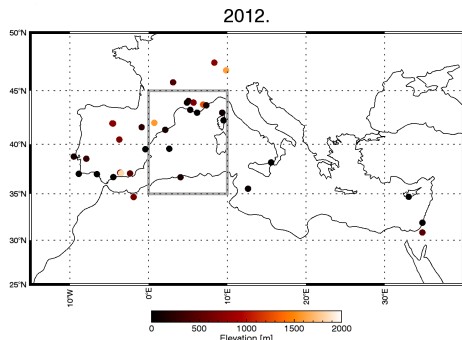

**Figure 5.** Positions of AERONET stations used in this study for the period of the TRAQA campaign in northern summer 2012. The grey box mark the region from which we considered SEVIRI data used in Fig. 3 and Table 1. The colours represent elevation of the stations as marked on the colorbar.

in Fig. 5. Timeseries plots for eight stations are presented in Fig. 6, and the statistics for all stations in Table 2. The stations in Fig. 6 are chosen to representatively cover the basin. The timeseries of the stations in the western part of the Mediterranean basin and in Spain are marked by the strong desert dust event, which was already discussed earlier. Stations in Spain registered the event before the stations in France, where it arrived a couple of days later. The duration of the event is well simulated

by both the direct model run and the assimilation model run in all stations, but the intensity is underestimated in the direct model run (Fig. 2). However, the assimilation model run matches well the outbreak intensity. The second, smaller desert dust event at the end of the TRAQA period is observed only at southern stations. Similarly, the assimilation model run corrects its intensity underestimated by the direct model run. The stations in the east, like in Lampedusa and Cyprus, were not influenced by these dust events. They are mostly influenced by sea salt aerosols, and the data assimilation also here has a very positive

impact. The assimilation model run, with only two MODIS overpasses per day, shows also improved hourly variations of AOD in these stations. These variations are not clearly visible in the model direct run, but they are present in AERONET data with similar amplitudes as in the assimilation model run. The statistics of all AERONET stations confirm the overall positive effect of assimilating MODIS data (Table 2).

The AERONET findings confirm those obtained by the comparison with SEVIRI observations. The scatterplot of all

AERONET observations (Fig. 7) reinforces the conclusion that the assimilation model run reduces the bias in the AOD field of the direct model run and significantly improves the statistical parameters.

### 6.4    In-situ aircraft concentration measurements

To further assess the performance of the assimilation model run we evaluate the impact of the AOD assimilation on aerosol properties other than AOD. To do this, we compare the modelled aerosol number concentrations with the aerosol concentra-

tions measured in-situ during the TRAQA period by the PCASP instrument. During the campaign, flights with the ATR-42



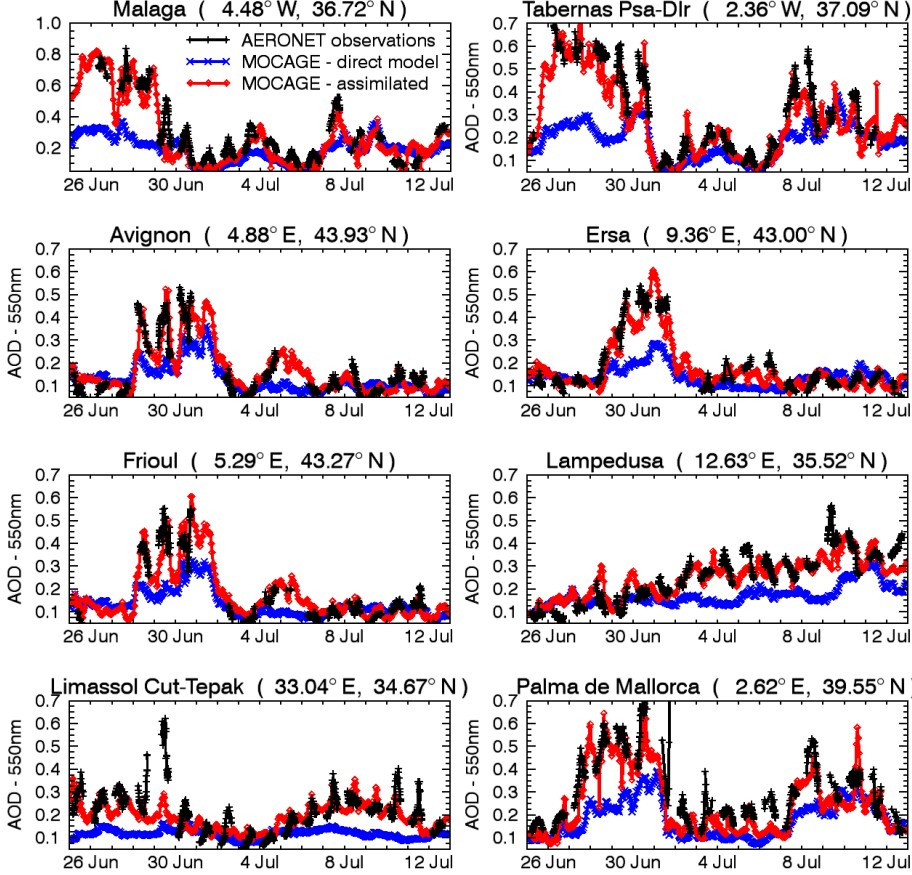

**Figure 6.** Time series of aerosol optical depth at 550 nm of the AERONET data (black line), the direct model (blue line) and the assimilation model run (red line) for the period of the TRAQA campaign from 25.06.2012 until 13.07.2012. The presented AERONET data are from eight stations: Malaga (ESP), Tabernas (ESP), Avignon (FRA), Ersa (FRA), Frioul (FRA), Lampedusa (ITA), Limassol (CYP), Palma de Mallorca (ESP). The location of a particular station is marked at the top of each panel. Correlation, bias and root mean square error for both the direct model and the assimilation model run as compared to the AERONET data are given in Table 2.

were conducted during 9 different days, carrying the PCASP instrument onboard. The flights passed over the whole western Mediterranean basin using Toulouse, Marseille and Corsica airports, and covered different meteorological and air quality conditions. Figure 8 presents three representative timeseries from these flights: flight A of 26.06.2012 from Corsica to Toulouse (Fig. 8a), flight B of 27.06.2012 from Marseille to Corsica and back to Toulouse (Fig. 8b), and flight C of 29.06.2012 from Corsica to Toulouse (Fig. 8c).

During Flight A (Fig. 8a), aerosol concentrations are rather low, except for the anthropogenic pollution around Toulouse measured at the flight end. The aircraft first visited the area of the Gulf of Genoa where, because of no available AOD observa-



**Table 2.** Correlation ($\rho$), absolute bias ($\Delta$) and root mean square error (RMSE) between AERONET observations and MOCAGE direct/assimilation run for the period of the TRAQA campaign between 25.06.2012 and 13.07.2012. Each of the stations in the Table is identified by its site name, latitude/longitude, station height, number of observations and above mentioned statistical parameters. AERONET site locations are also presented in Fig. 5.

| Station (Location) | Alt [m] | $N_{obs}$ | MOCAGE direct | | | MOCAGE assimilation | | |
|---|---|---|---|---|---|---|---|---|
| | | | $\rho$ | $\Delta$ | RMSE | $\rho$ | $\Delta$ | RMSE |
| Aubiere (FRA; 45.8°N, 3.1°E) | 423 | 225 | 0.506 | 0.041 | 0.110 | 0.859 | 0.024 | 0.074 |
| Autilla (ESP; 42.0°N, 4.6°W) | 873 | 685 | 0.769 | 0.012 | 0.085 | 0.882 | 0.008 | 0.051 |
| Avignon (FRA; 43.9°N, 4.9°E) | 32 | 846 | 0.851 | 0.024 | 0.087 | 0.896 | 0.005 | 0.055 |
| Barcelona (ESP; 41.4°N, 2.1°E) | 125 | 378 | 0.802 | 0.110 | 0.169 | 0.900 | 0.038 | 0.084 |
| Burjassot (ESP; 39.5°N, 0.4°W) | 30 | 488 | 0.681 | 0.132 | 0.191 | 0.815 | 0.055 | 0.119 |
| Cabo da Roca (PT; 38.8°N, 9.5°W) | 140 | 77 | 0.965 | 0.130 | 0.263 | 0.939 | 0.032 | 0.120 |
| Calern OCA (FRA; 43.7°N, 6.9°E) | 1270 | 509 | 0.784 | 0.013 | 0.093 | 0.905 | 0.009 | 0.052 |
| Carpentras (FRA; 44.1°N, 5.1°E) | 100 | 738 | 0.774 | 0.023 | 0.085 | 0.876 | 0.005 | 0.055 |
| Cerro Poyos (ESP; 37.1°N, 3.5°W) | 1830 | 193 | 0.632 | 0.034 | 0.061 | 0.670 | 0.055 | 0.074 |
| Davos (CH; 46.8°N, 9.8°E) | 1596 | 210 | 0.518 | 0.064 | 0.091 | 0.677 | 0.027 | 0.063 |
| Ersa (FRA; 43.0°N, 9.4°E) | 80 | 675 | 0.760 | 0.043 | 0.112 | 0.946 | 0.011 | 0.045 |
| Evora (PT; 38.6°N, 7.9°W) | 293 | 886 | 0.826 | 0.010 | 0.128 | 0.932 | 0.019 | 0.061 |
| Frioul (FRA; 43.3°N, 5.3°E) | 40 | 658 | 0.871 | 0.037 | 0.096 | 0.952 | 0.014 | 0.044 |
| Granada (ESP; 37.2°N, 3.6°W) | 680 | 883 | 0.677 | 0.041 | 0.129 | 0.930 | 0.003 | 0.057 |
| Huelva (ESP; 37.0°N, 6.6°W) | 25 | 1002 | 0.793 | 0.010 | 0.153 | 0.936 | 0.034 | 0.083 |
| Laegeren (CH; 47.5°N, 8.4°E) | 735 | 208 | 0.586 | 0.077 | 0.128 | 0.630 | 0.037 | 0.103 |
| Lampedusa (ITA; 35.5°N, 12.6°E) | 45 | 1058 | 0.573 | 0.084 | 0.124 | 0.845 | 0.006 | 0.061 |
| Limassol (CYP; 34.7°N, 33.0°E) | 22 | 978 | 0.338 | 0.080 | 0.115 | 0.640 | 0.003 | 0.067 |
| Madrid (ESP; 40.5°N, 3.7°W) | 680 | 904 | 0.731 | 0.011 | 0.097 | 0.878 | 0.004 | 0.058 |
| Malaga (ESP; 36.7°N, 4.5°W) | 40 | 786 | 0.702 | 0.101 | 0.173 | 0.910 | 0.047 | 0.088 |
| Messina (ITA; 38.2°N, 15.6°E) | 15 | 573 | 0.519 | 0.068 | 0.111 | 0.835 | 0.020 | 0.060 |
| Montsec (ESP; 42.1°N, 0.7°E) | 1574 | 528 | 0.662 | 0.016 | 0.078 | 0.892 | 0.009 | 0.044 |
| Nes Ziona (ISR; 31.9°N, 34.8°E) | 40 | 593 | 0.266 | 0.053 | 0.111 | 0.788 | 0.014 | 0.063 |
| OHP Observatoire (FRA; 43.9°N, 5.7°E) | 680 | 657 | 0.742 | 0.019 | 0.089 | 0.886 | 0.000 | 0.053 |
| Oujda (MAR; 34.7°N, 1.9°W) | 620 | 330 | 0.459 | 0.202 | 0.221 | 0.756 | 0.090 | 0.116 |
| Palencia (ESP; 42.0°N, 4.5°W) | 750 | 649 | 0.859 | 0.030 | 0.104 | 0.919 | 0.002 | 0.051 |
| Palma de Mallorca (ESP; 39.6°N, 2.6°E) | 10 | 797 | 0.754 | 0.129 | 0.163 | 0.888 | 0.048 | 0.084 |
| Porquerolles (FRA; 43.0°N, 6.2°E) | 22 | 637 | 0.805 | 0.005 | 0.071 | 0.923 | 0.020 | 0.044 |
| Sagres (PT; 37.0°N, 8.9°W) | 26 | 405 | 0.901 | 0.017 | 0.197 | 0.958 | 0.023 | 0.088 |
| Sede Boker (ISR; 30.9°N, 34.8°E) | 480 | 950 | 0.240 | 0.009 | 0.080 | 0.552 | 0.065 | 0.095 |
| San Giuliano (FRA; 42.3°N, 9.5°E) | 10 | 768 | 0.675 | 0.084 | 0.137 | 0.908 | 0.066 | 0.089 |
| Tabernas (ESP; 37.1°N, 2.4°W) | 500 | 740 | 0.754 | 0.129 | 0.184 | 0.927 | 0.038 | 0.078 |
| Tizi Ouzou (DZA; 36.7°N, 4.1°E) | 133 | 241 | 0.686 | 0.195 | 0.203 | 0.764 | 0.079 | 0.095 |
| Villefranche (FRA; 43.7°N, 7.3°E) | 130 | 480 | 0.707 | 0.064 | 0.113 | 0.873 | 0.033 | 0.067 |
| Zaragoza (ESP; 41.6°N, 0.9°W) | 250 | 916 | 0.722 | 0.053 | 0.100 | 0.816 | 0.040 | 0.073 |
| All sites | | 29840 | 0.740 | 0.050 | 0.115 | 0.883 | 0.006 | 0.070 |



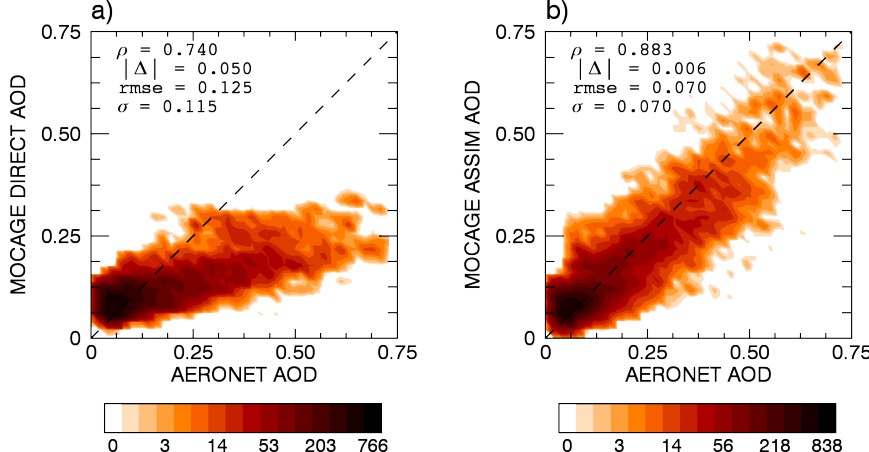

**Figure 7.** Scatterplots of aerosol optical depths (where colours represent the number of points) from the independent observation dataset (AERONET) and the simulations: the direct model run **(a)** and the assimilation model run **(b)**. In each panel, correlation ($\rho$), absolute bias ($\Delta$), root mean square error (RMSE) and standard deviation ($\sigma$) are noted. The included data correspond to the period of the TRAQA campaign from 25.06.2012 until 13.07.2012., and covers all stations presented in Fig. 5.

tions, the direct model run and the assimilation model run show the same aerosol concentrations. The variability in that part of the flight is well simulated with slightly higher modelled aerosol concentrations at these heights than what is measured. Later, on the way to Toulouse, with more available satellite observations, the assimilation model run lowers AOD and approaches the measured concentration values. With situations of no observations or with sparse ones, data assimilation is not able to have a

5 major effect.

Following the path of Flight B (Fig. 8b) we see again rather clean aerosol conditions. The modelled and assimilated curves differ, and the assimilation has an effect on the shape of the timeseries curve, but it does not improve noticeably the field compared to the measurements. The result that the assimilation changes the model, but without a clear improvement, could be due to different factors. This could happen if the simulated shape of the aerosol vertical profile somehow differs from the

10 measured one. Since we assimilate the column integrated quantity which does not contain the profile shape information, the model and measurements at the certain height will not match if the modelled profile shape is far from the real one, although the AOD values could match well. A similar impact could arise if the modelled mixture of aerosol types and the size distribution are different from the real one. The difference in the sizes would be easily noticeable in the aerosol number concentration, even if the modelled and observed AOD values correspond well. The third factor that could contribute to the difference between the

15 modelled and the assimilated curve is a possible declining effect with time of previous assimilation cycles on the assimilated curve.

During Flight C (Fig. 8c), the aircraft flew directly through the desert dust plume. The concentrations are elevated over a wide range of heights. The assimilation model run significantly improves the aerosol number concentration, by having it close





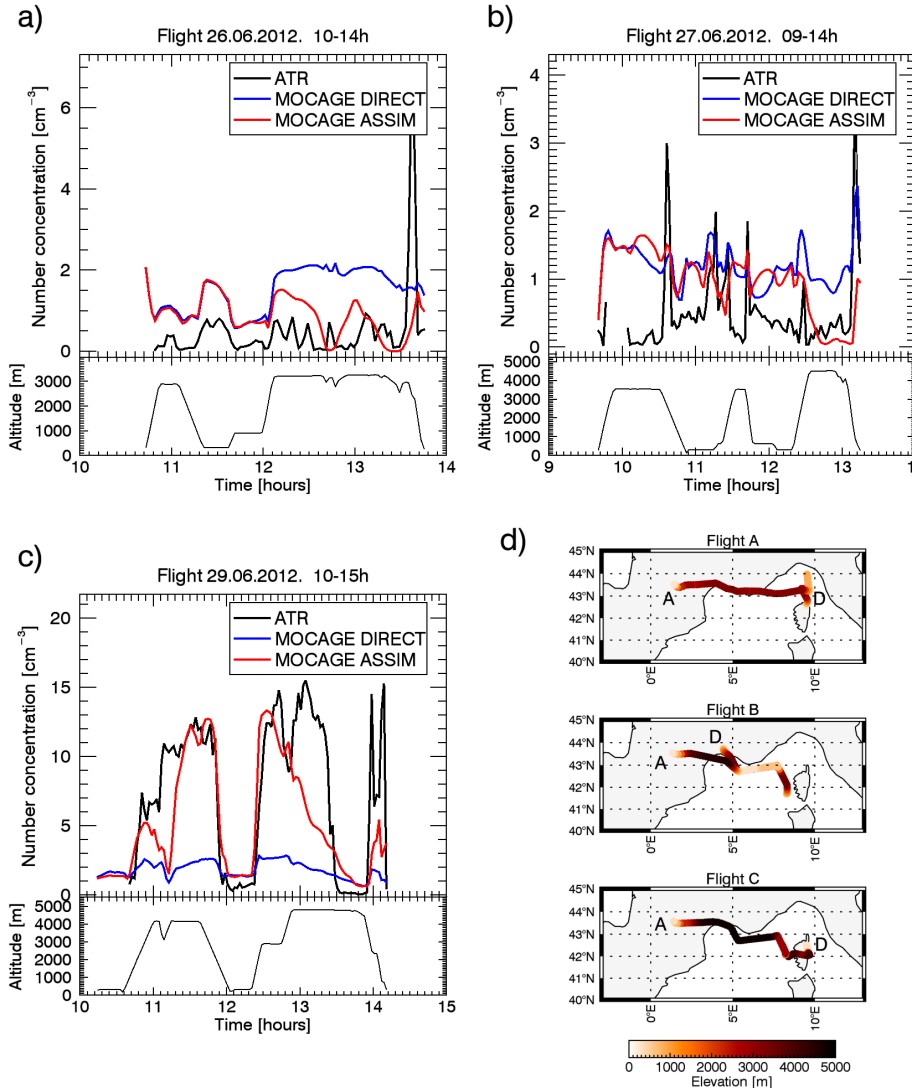

**Figure 8.** Aerosol number concentration [cm$^{-3}$] from the PCASP instrument onboard of the ATR aircraft (black line) compared with the direct model run (blue line) and the assimilation model run (red line) for three different flights: Flight A on 26.06.2012 **(a)**, Flight B on 27.06.2012 **(b)**, and Flight C on 29.06.2012 **(c)**. The altitude of the aircraft is also given for all three flights. Also, the maps of flight tracks are presented **(d)**, as well as the points of departure (D) and arrival (A) for each flight. The colours of the tracks represent the altitude of the aircraft during the flight with values defined in the colorbar. The aerosols are considered in the size range from 1 μm to 2.5 μm.

to the measured ones for most of the flight path within the plume. If satellite measurements are accurate, concentrations at one height after assimilation can closely correspond to measured ones only if the shape of the vertical profile is well simulated in the direct model run. To further explore this, we compare the modelled and the measured vertical profiles follows.




## 6.5 In-situ balloon concentration measurements

During TRAQA, LOAC flew on three balloons, all launched from Martigues, near Marseille (FRA). Two flights on 29.06.2012, and one on 06.07.2012 are presented in Fig. 9. The first two flights flew through the desert dust plume. The path of the second LOAC flight is near the path of the aircraft Flight C (Fig. 8c), which allows us to directly compare the two measurements. The total horizontal motion of LOAC is fairly small $\leq 15\,\mathrm{km}$. Therefore, we will assume that LOAC measurements represent the aerosol vertical profile above the launch place.

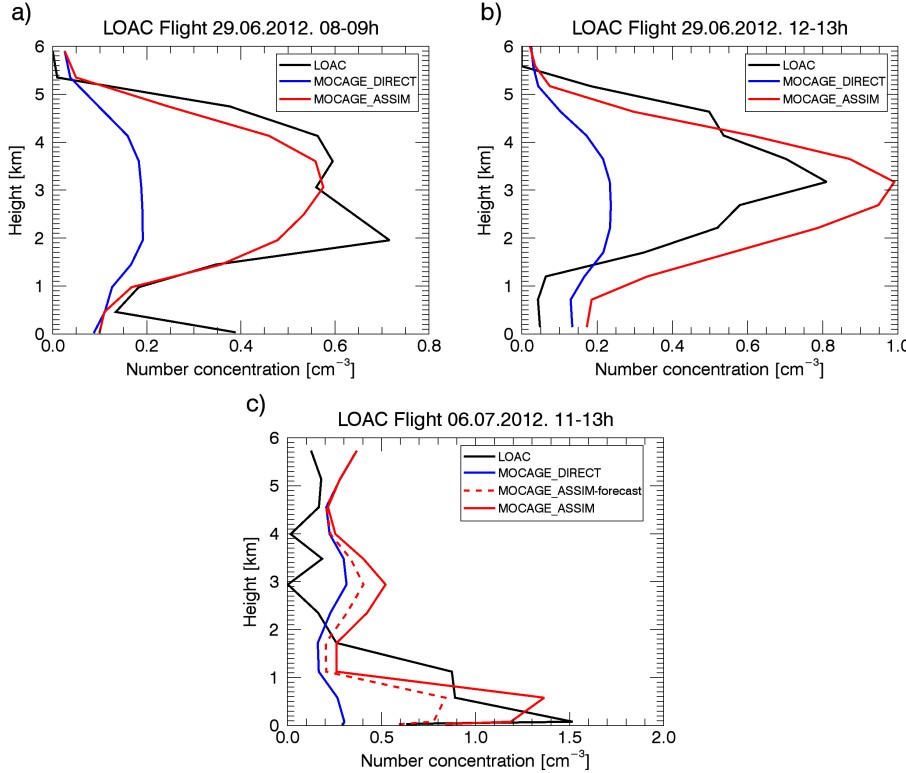

**Figure 9.** Aerosol number concentration $[\mathrm{cm}^{-3}]$ from the LOAC instrument onboard on meteorological sounding balloons. The presented flights are performed: in the morning of 29.06.2012 **(a)**, at noon of 29.06.2012 **(b)**, and on 06.07.2012 **(c)**. LOAC measurements (black line) are compared with the direct model run (blue line) and the assimilation model run (red line). For the third flight, we also present the one hour direct model forecast started from the assimilated conditions of one hour before (dashed red line). The aerosols are considered in the size range from $2.5\,\mu\mathrm{m}$ to $100\,\mu\mathrm{m}$.

The first two flights (Figs. 9a ans 9b) are launched at two different times of the same day, in the morning and at noon, but they flew through the same desert dust plume. In both cases, the assimilation model run matches very closely the measurements. It simulates well both the shape of the profile and the aerosol number concentration. The direct model run simulates well





the shape of the vertical profile, but it underestimates the aerosol concentration in the plume which the assimilation corrects. Sometimes, the generalized multiplicative change of the aerosol profile in the assimilation can produce unsatisfactory effects in some layers. For the second flight, although the concentrations in the plume are hugely improved, near the surface the increase of aerosols lead to a larger overestimation in the model.

5  LOAC measurements acquired during the second flight are colocated with the aircraft measurements (Fig. 8c). They match well with the assimilation model run profile which confirms the already discussed interpretation: the aerosol concentration in the assimilation run at a certain height can be correct only if the profile shape is well simulated in the direct model run.

The third LAOC flight (Fig. 9c) measured moderate aerosol concentrations coinciding with an air pollution episode. The assimilation model run matches well with the measured concentrations. The direct model run underestimates the concentrations 10 only in lower levels. But, when compared with the assimilation model run, the assimilation changes the aerosol vertical profile significantly: the concentrations are increased much more in the lower levels, while in higher levels the change is less important. In this case, the different shape of the profile in the direct model run and the assimilation model run is a result of the continuous multiday assimilation of AOD over many assimilation cycles, and the mixing of the aerosols coming from different levels and regions where they were already assimilated (or not) in previous assimilation cycles. This demonstrates that the continuous 15 assimilation of good quality AOD observations can correct a shape of the aerosol vertical profile, although a single AOD assimilation cycle can only expand or shrink the profile shape (as the AOD observations do not contain the information on the vertical). For the profile in Fig. 9c, by comparing the forecast and the analysis of the same assimilation window, we see that the single AOD assimilation cycle expands the profile but does not change its shape, what multiple cycles does.

## 6.6 The profile evolution

20  The profile evolution in the continuous multiday assimilation run is further explored in Fig. 10. We follow the desert dust plume over the course of one week from 25 June to 2 July 2012 from its sources in Africa till its weakening and dissolution in the Mediterranean sea (Fig. 10c). The aerosols at the different layers are carried by the winds at different velocities and directions, and, as particles, they undergo different physical processes (dry and wet deposition, sedimentation, etc). Therefore, we cannot follow the dust plume by following an air parcel. Instead, to track the plume we use a criterion based on the high values of 25 AOD. The plume, after its emission heads west and passes over a couple of other dust sources. Near the Canary islands the plume turns north-east towards the Mediterranean basin under the influence of a low-pressure system centered near the British Isles. MODIS directly observes the plume each day during the considered period (MODIS passes are marked by triangles in Fig. 10b), and these observations in the assimilation model run have a considerable impact on the plume. When comparing the direct model run (Fig. 10a) and the assimilation model run (Fig. 10b), the most obvious assimilation impact is the change of the 30 intensity of the plume in the first part of the trajectory. But, also the profile shape evolves considerably: in different moments the plume maxima are shifting their peak heights, and the different layers are changing their relative densities.

There are different effects that the continuous assimilation can have on the profile shape. While following the plume based on the AOD values, different air masses enter and exit our trajectory based on their different transport velocities and directions. Thus, aerosols that are assimilated at different places later can be gathered in one vertical profile. In Figs. 10a and 10b, this



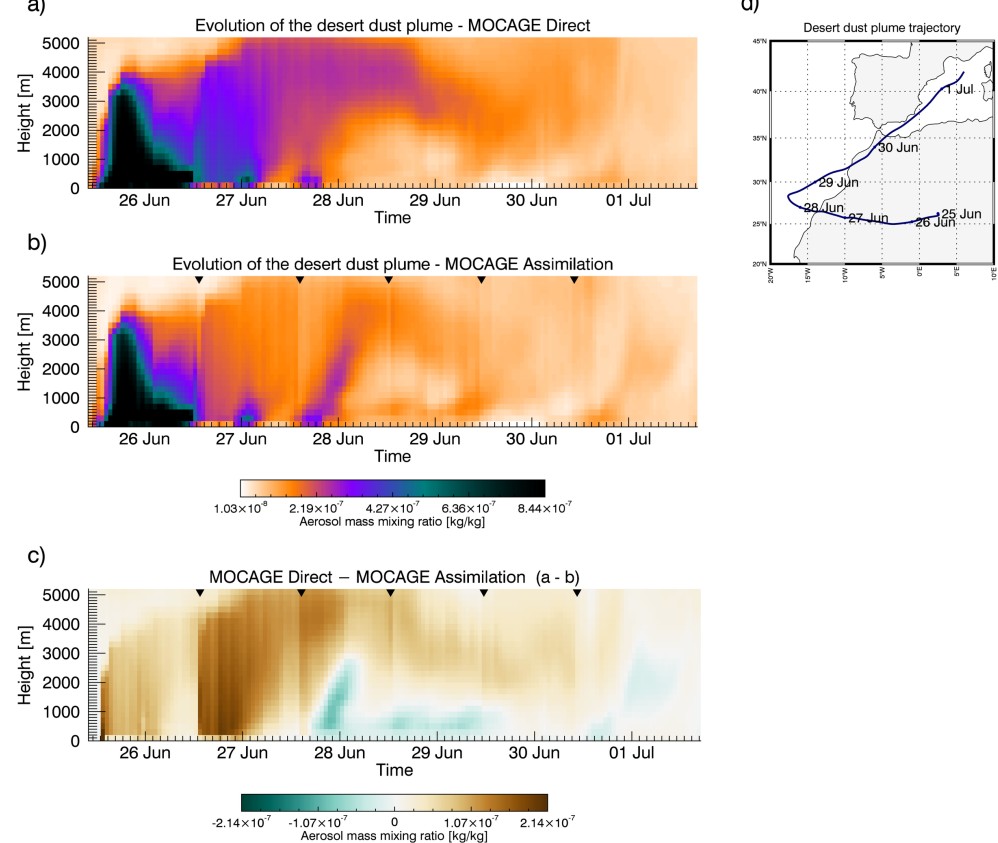

**Figure 10.** The evolution of the aerosol vertical profile over the course of one week from 25 June to 2 July 2012 in the direct model run (**a**) and the assimilation model run (**b**) and the difference between the direct and the assimilated model run (**c**). The units are in mass mixing ratios with the values represented by colours identified in colorbars. In figure (**b**) the passes of the MODIS over the plume are marked by the triangles. The desert dust plume is followed from its sources in Africa till its weakening and dissolution in the Mediterranean sea (**d**).

effect on the plume is noticeable around 1 July in the northern part of the basin where wind velocities and directions vary with altitude and where airmass mixing is pronounced. Another effect of the multicycle assimilation is amplifying or reducing of relative differences between layers in a profile. For example, in Fig. 10, the assimilation on 26 and 27 June lowers the intensity of the plume in the higher layers, and the subsequent assimilation cycles amplify aerosol concentrations in the lower layers. On 30 June, the assimilation increases even more the relative difference between the layers. As a result, aerosol mixing ratios are larger in lower layers than in higher layers, while in the direct model run this is the opposite.



## 7 Discussion

Results presented in this paper showed that the AOD data assimilation is an efficient technique to improve modelled aerosol fields. Assimilated fields have better statistical performances than the direct model run in comparison with the assimilated observations, and with independent AOD observations and in-situ measurements. The uncertainties in the direct model in the case of a desert dust outbreak come primarily from the uncertainties in the desert dust emissions. The dust emission into the atmosphere is a threshold process, which is very sensitive to uncertainties in the wind field. The small changes in wind can produce significant differences in the emitted quantities. The AOD assimilation proved to be a technique capable of reducing the effects of such uncertainties in the modelled aerosol fields.

In doing so, our assimilation system showed to be more efficient in lowering overestimated AOD values in the model than increasing underestimated values. This can be seen in Fig. 1 and it is directly related on how matrices $\mathbf{B}$ and $\mathbf{R}$ are defined in the experiment. By defining variances as the percentage of modelled and observed quantities, and making this factor two times smaller for observations, we penalized the high AOD values in the model. This directly affects the analyses, and later reflects in the forecasts.

To regulate this feature, one of the possible and the simplest (ad-hoc) approaches would be to limit the observation error in the matrix $\mathbf{R}$ up to a fixed value, which would give more weight to observations in the case of high observed AOD. This would have a partial effect, influencing only observations above a certain AOD. Another approach would be to try to define the matrix $\mathbf{B}$ differently. Previous studies show that a rigorously defined matrix $\mathbf{B}$ can slightly improve the analysis quality (Kahnert, 2011; Massart et al., 2012). In MOCAGE-Valentina, in the framework of the MACC (Monitoring Atmospheric Composition and Climate) project, the influence of different matrices $\mathbf{B}$ was assessed for the case of ozone assimilation (Jaumouillé et al., 2012). One of the approaches was the percentage method used in our experiment. The second approach was the monthly *a posteriori* diagnostics (Desroziers et al., 2005) computed from the data of a month before, and it is adapted for operational purposes since the data from the past is readily available. The third approach is to calculate diagnostics from an ensemble of runs with perturbed emissions with homogeneous or heterogeneous correlation length scales. The main conclusion is that all methods significantly improve the modelled field, and that relative differences between different methods are small compared to the rate of improvement by assimilation. For aerosols, the transport processes are more important than for ozone, but the work of Jaumouillé et al. (2012) could give an idea of what to expect in the model from redefining the construction of the matrices $\mathbf{B}$ and $\mathbf{R}$. Moreover, it should be kept in mind that using a dense observation field, like our MODIS superobservation field, limits the effects of the spatial propagation of the increment. This makes the covariances of the matrix $\mathbf{B}$ less important than in the case of sparse observations.

In our system, the lack of secondary aerosols is presumed to have an influence on assimilation performance. This could lead to an underestimation of the direct model AOD in the regions where the secondary aerosols have an important influence. During the TRAQA period, the primary aerosols had a dominant effect on the aerosol field, mainly because of the two desert dust events that occurred during the campaign, and this was favourable for the evaluation of our system. In the direct model, differences between the model and the observations appear because of different model uncertainties, including simplified and



neglected processes. But, the differences do not have a constant or cumulative nature; the model sometimes overestimates or underestimates AOD. To take into account these uncertainties in the assimilation process, we defined the variances in the matrix **B** in such a way that it allows a margin for the model errors (Talagrand, 2003). The developments of an inorganic secondary aerosol module in MOCAGE are carried in parallel with the developments on the AOD assimilation system (Guth
et al., 2015). This has an beneficial effect in the model and consequently, is expected to improve the analysis after its inclusion in the assimilation module. To take into account the model uncertainties there are also other possibilities. One would be to add an additional term in the cost function where we would describe the errors of the model evolution. This method can be used in the 4D-Var systems and, besides the implementation of the 4D-Var method, it demands additional computational resources in assimilation. Also, it is difficult to define the model error covariance matrix (Tréemolet, 2006). Another possibility would be
to apply techniques of bias correction (e.g. Dee and Uppala, 2008).

The impact of the AOD assimilation on the model found in our study is coherent with findings of other studies (Zhang et al., 2008; Liu et al., 2011; Schutgens et al., 2010). Our approach is similar to the approach used by Benedetti et al. (2009) since we choose the same control variable. The difference in our systems is the number of bins to which the increment is repartitioned (11 bins for five species in Benedetti et al. (2009) and 24 bins for four species in our system). Benedetti et al.
(2009) derived the matrix **B** using the NMC (National Meteorological Center) method (Parrish and Derber, 1992). Satellite AOD errors are defined for retrievals over water by using a multiregression formula, and for retrievals over land by using the percentage approach with defining a minimal possible error. Their 4D-Var analysis results showed qualitatively a very similar impact of assimilation as in this study.

The size distribution in the model does not change significantly by the assimilation (not shown). In our approach the *a*
*posteriori* influence on the bins is fixed, and the subsequent model evolution does not bring important differences in the mass concentration size distribution between the direct model run and the assimilation model run. Since the measured size distribution is a bulk aerosol size distribution, to compare it with the model we consider that in the dust plume only the desert dust particles are measured. Still, the overlap between the model and the PCASP instrument size range is only two model bins, and no conclusion could be made when comparing the modelled and measured size distributions. The quantity
of aerosols is better represented in the assimilated size distribution than in the direct model size distribution, and this point is already examined in the previous section and figures. Still, the agreement of the model with colocated PCASP and LOAC measurements that see aerosols in different size ranges points to a good representation of the accumulation and the coarse mode of the dust aerosols in the model.

We assimilated the MODIS data that has two overpasses per day during daytime. Satellite data with higher temporal res-
olution exist. SEVIRI data with a temporal resolution of 15 minutes was used as independent data to evaluate the results. Assimilating such data could further improve the agreement between observations and the assimilation model run, but considering the moderate temporal variability of AOD fields we would not expect a substantial improvement. Also, the SEVIRI AOD products are less accurate compared to MODIS. The TRAQA period used in our experiment is in northern summer with a good likelihood of having a cloud-free field, and two overpasses per day were able to cover a significant part of the control domain





each day. Possibly, a higher temporal resolution data for assimilation could have a stronger effect on the model, especially during the winter-time.

## 8  Summary and conclusion

In this study we present the development and validation of the MOCAGE-Valentina system for assimilating aerosol optical depth. Our system assimilates aerosol optical depth (AOD) with the 3D-FGAT method and uses the total 3D aerosol concentration as a control variable. We examined how 2D AOD observations in a continuous multicycle assimilation can influence the model aerosol representation, including the vertical aerosol profile. We used accurate in-situ aircraft and balloon measurements plus other remotely-sensed data to provide independent validation of the impact of the assimilation. The MODIS L2 data is assimilated with the model resolution of about 0.2 degrees and a one hour assimilation cycle over the region covering North Africa, the Mediterranean basin, and South Europe for the period of the TRAQA campaign in the northern summer of 2012.

The assimilated model shows greatly improved aerosol representation compared to the independently observed data sets, including the 3D distributions. The comparison with SEVIRI and AERONET AOD observations, as independent datasets, confirmed the significant positive effect of the AOD assimilation to the model. For example, the comparison with AERONET data showed that the assimilation decreased the bias in AOD (from 0.050 to 0.006) and increased the correlation (from 0.74 to 0.88).

The TRAQA campaign provided independent 3D data on aerosol concentration and the size distribution. The assimilation sometimes improved the modelled fields and sometimes had little effect. The best results as expected occurred when the shape of the vertical profile is correctly simulated by the direct (unassimilated) model. The shape of the aerosol vertical profiles does not change during one assimilation cycle because AOD observations do not contain any vertical information, but the profile shape can change and be improved by the AOD assimilation because different parts of the column can be carried by winds from different directions. The AOD assimilation can also impact aerosol size and type for the same reason, but this was not evident in this experiment. The AOD assimilation proved to be a very efficient technique to improve the model forecast of bulk aerosols and a powerful tool for producing reanalyses or studying past events.

As of an outlook of further developments, the next steps will consist of improving the system performance and broadening its capabilities. For example, it could be advantageous to assimilate observations from different instruments at the same time. The AOD observations from space are available from various instruments located on different satellites, which can provide different spatial and temporal coverage and resolutions. Combining complementary data from different instruments could improve the system performance.

Also, the same or even bigger positive effect of the assimilation could be expected in the case of other strong aerosol events, like biomass burning or a volcanic ash plume, where the model emission uncertainties are often even larger than in the case of desert dust plumes.

By assimilating AOD observations at several wavelengths, we can get information on aerosol size. If aerosol absorption can be measured then we can discriminate between carbonaceous and other aerosols. Then, with this information we could modify



the size distribution and aerosol bin distribution in the model. To achieve this in the system, it would be necessary to study the relationship and sensitivity between the size and bin distribution in MOCAGE and the aerosol Ångstrom exponent obtained from multi-wavelength measurements.

If we want to introduce direct informations of the vertical profile from observations into the model, we would need to as-
5 similate another type of observations. Lidar observations, ground- or space based are an obvious choice. The control variable defined as the 3D total concentration is also well adapted for the assimilation of lidar profiles. This facilitates the implementation of the lidar assimilation into the system, and in the longer term also makes feasible a simultaneous assimilation of the AOD and lidar profiles as possibly complimentary datasets.

*Acknowledgements.* This work has been funded by Centre National de Recherches Météorologiques (CNRM-GAME) of Météo-France
and Centre National de la Recherche Scientifique (CNRS). The authors would like to thank Jean-Luc Attié, the TRAQA PI, and all TRAQA/ChArMEx collaborators for producing and providing the data used in this study. TRAQA was funded by ADEME/PRIMEQUAL and MISTRALS/ChArMEx programmes and Observatoire Midi-Pyrénées. We acknowledge the AERONET PIs and their staff for establishing and maintaining the sites used in this investigation. We also acknowledge the Global Fire Emission Database project and Lamarque et al. (2010) for the biomass burning and carbonaceous aerosol emissions, the MODIS mission team and scientists for the production of the
MODIS data, as well as Thieuleux et al. (2005) and the ICARE data center for developing and producing the SEVIRI-retrieved aerosol data that we used in this study.



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
