# Peer review of "Aerosol data assimilation in the chemical-transport model MOCAGE during the TRAQA/ChArMEx campaign: Aerosol optical depth"

_Atmospheric Measurement Techniques, 2016_

## Referee Comment (RC1) · Anonymous Referee #2 · 4 May 2016

**Aerosol data assimilation in the chemical-transport model MOCAGE during the TRAQA/ChArMEx campaign: Aerosol optical depth**

Bojan Sic, Laaziz El Amraoui, Andrea Piacentini, Virginie Marécal, Emanuele Emili, Daniel Cariolle, Michael Prather, and Jean-Luc Attié

This paper describes a data assimilation system for aerosol applied to a regional model. While the DA system itself is not novel, the interest of the paper lies in the verification of DA results with observations very different from those used in the data assimilation. Assimilated observations are remotely sensed AOT from MODIS but part of the verification is done with flight campaign measurements of particle number densities. Sic et al show that although they assimilate only column-integrated optical properties (AOT), even particle number density profiles can be improved. This is a very interesting result.

Sic et al. go on to explain this in terms of how the modelled profile shape changes in subsequent assimilation cycles. Although this is not really surprising to specialists in aerosol data assimilation, it can be hoped that this paper will help affect a better understanding in a more general audience who often have trouble accepting that data assimilation of 2D properties can improve 3D distributions.

This point might be brought across even stronger with some additional analyses. Some suggestions are
   - Trajectory analyses of selected air parcels identified during previous assimilation cycles,
   - A data denial experiment where the DeepBlue observations are removed (and so observations close to dust sources are missing),
   - Evaluation against LIDAR data, e.g. Earlinet.
That said, the paper is acceptable for publication as it is (after minor changes suggested below).

Specific comments:

Abstract:

L 14: "reduced the bias in the AOD (from 0.050 to 0.006 ) and increased the correlation (from 0.74 to 0.88 )": please state for which dataset (MODIS, SEVIRI, AERONET). Also, include a standard deviation or RMS error. If any spatio-temporal averaging was done beforehand, specify so now.

Introduction:

P 2, L 3: Neither Textor nor Lee consider efforts to improve models. Textor evaluates an ensemble of models, while Lee developed a methodology to better understand a single model. True, all of this with a goal to finally improving the

models but neither of these papers actually gets there. There is a vast literature out there about model evaluation and development, and more (and more relevant) references should be included.

P2, L 6: "diversity of parameterizations": there are many causes (e.g. dynamical cores, emissions, parameter uncertainty) for model diversity, parametrisation of processes is only one of them.

P2, L19: Please include at least the following: Collins et al JGR 2001 (OI), Weaver et al JAS 2005, Generoso et al JGR 2007 (3D-VAR), Zangh et al JGR 2008 (3D-VAR), Yumimoto et al. ACP 2008 (4D-VAR), Sekiyama et al. ACP 2010 (EnKF), Schwartz et al. 2012 JGR (3D-VAR), Saide et al ACP 2013 (3D-VAR), Rubin et al ACP 2016 (EnKF)

P3, L26: "The particle size distribution for each type is divided into 6 size bins, characterized by the particle average diameter and mass. Each aerosol bin is then treated as a passive tracer: aerosols are emitted, transported and removed from the atmosphere, however there are no transformations or chemical reactions between aerosol types, between size bins or with gases." To put it differently: a 2-moment scheme (mass & numbers) is used to describe aerosols in each bin but no coagulation or condensation processes are modeled? If these latter processes are not considered, wouldn't a simple mass scheme do just as well?

P4, L7: "We use daily BB emissions for better synoptic forecasts, which is not possible with the monthly mean emissions of Lamarque et al. (2010)." But this begs the question how a diurnal cycle in emissions would affect your standard model run (and your DA results for that matter). Emission datasets with (imposed) diurnal cycles for anthropogenic emissions exist for the European domain, e.g. TNO-INERIS. Why were such datasets not used?

P5, L2: "The model is then run over a cycle length (1 hour) to obtain the analysed trajectory." I could not find any clear statement over which domain observations are assimilated. Sect. 4 seems to suggest only over MEDIO2. But why then a cycle of 1 hour? New MODIS observations will happen only twice every day (at best). Since the cycle is only 1hr, isn't 3D-FGAT very similar to standard 3D-VAR?

P5, L13: Reformat "Courtier et al. (1994)"

P6, L3: "nor does it need to contain the inter-bin covariances"

P6, L6: "could be weighted by 5  different quantities, like number or mass concentration, or extinction coefficient". I disagree. If the innovation is zero (i.e. forecast AOT agrees with observations), it makes sense that the relative distribution of aerosol species does not change. This limits how you can redistribute aerosol. I also don't understand how number and masses would make a difference in your system. From an earlier description, I gather that masses and numbers in each bin are closely related (since there are no inter-bin processes or gaseous condensations).

P6, L20: "also by taking into account the hygroscopicity of sea salt aerosols" Surely the hygroscopicity of sulfate and organic carbons is considered as well? This is part of the OPAC (by the way, I believe the authors may be mistaken the climatology GADS for the aerosol optical properties database OPAC)

P6, L27 and further: I am not sure what this adds to the paper. Consider removing it or clarifying its importance.

P7, L16: "defines the observation and representativeness errors. These errors are considered to be non-correlated, which means that all non-diagonal members (covariances) in the matrix R are zero". This is an assumption that is unlikely to be true, especially given that your super-observations are aggregates over 0.2 by 0.2 degrees (a very short distance). At the very least you need to state this, better yet if you can provide justification for your assumption.

P8, L1: "As the optimal parameters, we estimated that the percentage for the errors of the model should be twice as large as for the observations (24% and 12% respectively)." I don't understand what the 24 and 12% refer to. R should be based on realistic assessment of the observational error (including representativeness error), independent of the data assimilation. So is 12% a typical observational error? But it seems estimated from Eq 13, so it is not independent from the data assimilation system.

P8, L10: "The implemented horizontal correlation length is 0.4." This is based on other studies or just an educated guess?

P8, Section 4: It appears that outside MEDIO2 no observations are assimilated? Why not? Wouldn't you want to constrain inflows across the MEDIO2 boundaries?

P8, L20: MODIS C5 is known to have systematic biases, e.g. due to windspeed and cloud cover over ocean but also over land. This has greatly improved in C6 so why did you not use this? Also, these error estimates are very optimistic (see Fig 20 in Schutgens et al AMT 2013). Even C6 has higher error estimates.

P8, L25: Earlier the authors mention representativeness errors. Do you assume these are essentially zero because of the creation of super-observations? But you will seldom have grid-filling MODIS observations so representativeness may still be an issue. Also, over distances of less than 50km aerosol is usually strongly correlated. As a result, retrieval errors and representativeness errors may also be strongly correlated. Yet you assume they are not. Please discuss this.

P9, L5: What are typical SEVIRI AOD errors over ocean? Did you consider assimilating these data (which have a much higher time resolution than MODIS)?

P9, L8: 3m and 5m should likely be 3km and 5km.

P9, L12: Why did the authors use AERONET L1.5 data? Since their experiment is

from 2012, L2 data are available and generally far superior in cloud screening. It seems they do not consider AE as a observation for evaluation. Why not? Have they considered using AERONET inversion data for evaluation? SSA measurements would allow for an interesting evaluation of aerosol speciation.

P9 & 10 PCASP and LOAC: is anything known of measurement errors for these instruments (after the time-averaging)?

P10, L12: Is there a separate spin-up of the assimilation as well? Or do the authors start evaluating results after the very first assimilation cycle?

P10, L13: "The assimilation cycles in the experiment have a length of one hour." I guess most observations are made near the same time so why use a one hour cycle?

P10, L12: "(directly forecasted" Is this the direct experiment or the forecast that is part of the assimilation experiment?
P10, L22: This suggests that biases are small (RMS error ~ standard deviation) and that suggests you are comparing forecast and analysis after some spin-up of the assimilation system?

P12, L3: "MODIS overpasses each point twice" Undoubtably the authors are aware there are two MODIS instruments, each passes over a point once a day.

P 13, Fig 4: How do the authors interpret the difference in direct model run for MODIS (Fig 1) and SEVIRI (Fig 4). The first one seems unbiased but in the second one the model is clearly biased low. Is this because Fig 1 really compares the forecast and not the direct model run? If so, how does the direct model compare against MODIS and does it look similar as Fig 4?

P 18 Fig 8: I find this very interesting and a major result of this paper. Maybe panel d) can be changed to include assimilated observations during the day of the flight? It would help understand what information was available and what not.

P20, L3: "Sometimes, the generalized multiplicative change of the aerosol profile in the assimilation can produce unsatisfactory effects in some layers. For the second flight, although the concentrations in the plume are hugely improved, near the surface the increase of aerosols lead to a larger overestimation in the model." To me it seems these two sentences make more sense if their order is reversed.

P20, L14: "This demonstrates that the continuous assimilation of good quality AOD observations can correct a shape of the aerosol vertical profile, although a single AOD assimilation cycle can only expand or shrink the profile shape (as the AOD observations do not contain the information on the vertical). For the profile in Fig. 9c, by comparing the forecast and the analysis of the same assimilation window, we see that the single AOD assimilation cycle expands the profile but does not change its shape, what multiple cycles does." I agree with this

assessment and think it shows nicely the power of data assimilation. However, the authors may want to reconsider earlier statements like P20, L6 "the aerosol concentration in the assimilation run at a certain height can be correct *only* if the profile shape is well simulated in the direct model run." which is far more restrictive.

P20, L18: "multiple cycles" it may be useful to point out to readers that it is not just the multiple assimilation cycles that improve the profile  but also assimilation throughout a spatial domain. So dust originating from different regions and transported at different levels is adjusted in previous cycles before it ends up in the main plume. One question is then how important the DeepBlue observations are to your assimilation. I expect quite important because they contain information closest to individual dust source regions.

P20, L23: "Therefore, we cannot follow the dust plume by following an air parcel." I think I understand this sentence but the dust is still advected by the winds: following air parcels at different heights should give you the dust plume or all Lagrangian analyses are useless (HYSPLIT or FLEXPART).

P21, Fig 10 (panel d): is it possible to somehow plot Lagrangian trajectories as well to show how different air parcels move before contributing to the plume? This would be a very strong visual support for the nice analysis in 6.6

P22, L9: "In doing so, our assimilation system showed to be more efficient in lowering overestimated AOD values in the model than increasing underestimated values" This has been shown in several other assimilation studies as well. It's due to model errors being defined relative to the model forecast which is unavoidable if you assume the model to be unbiased. I expect your solution (in L14) would not work: if the model substantially underestimates actual AOD (i.e. outside the model's error envelope), adjusting observational error will not help (not to mention that it is close to fudging).

P23, L12: Also, Benedetti uses a 4D-VAR method not 3D-FGAT

P23, L19-28: This is the first time that mention is made of the very different size ranges of model and observations. The authors should state the model range (I could not find it in their paper) and how they dealt with the discrepancy (I assume they only compared number concentrations in overlapping bins). Ideally, this should be described in resp. the model and observation section but not here in the discussion. Instead of this long paragraph a single sentence that there was insufficient information to compare size distributions should be sufficient.

P24,L6: Instead of 'influence' I suggest 'improve'.

---

## Referee Comment (RC2) · Anonymous Referee #3 · 22 Jul 2016

**Referee comment for**

**Aerosol data assimilation in the chemical-transport model MOCAGE during the TRAQA/ChArMEx campaign: Aerosol optical depth (Sič et al., A.M.T.D., 2016)**

**General comments**

This is a well-designed and well-written paper describing a Data Assimilation (DA) system built on top of a chemical-transport model, its application to aerosol optical depth (AOD), and its performance through comparisons with assimilated and independent observations. As noted by the first reviewer already, the main achievement is the successful comparison with vertical profiles of concentrations obtained from in-situ balloon measurements. This shows, in a real-life case, how the assimilation of two-dimensional optical depths can improve the vertical distribution of aerosols. This positive result is convincingly attributed to the accumulation over time of assimilation increments peaking in layers which are correctly located by the model at emission time, and transported afterwards.

Section 7 needs revision to become clearer and more consistent with the results chosen for publication in section 6 (see specific comments 10-13). But this study has only one real weakness in my opinion: an insufficient discussion of the issue of speciation between different aerosol types - both in the description of the DA system and in the results section. One would have expected that the same mechanism improving vertical distribution would also improve aerosol speciation, especially in the present case study of a desert dust event originating in the Sahara. Yet it is simply stated in the conclusion that "*AOD assimilation can also impact aerosol size and type for the same reason, but this was not evident in this experiment*". It is important to fully document this negative result as well, especially w.r.t. aerosol type, in order to indicate the direction for future progress. The improvement of aerosol speciation thanks to DA of satellite AOD would indeed be a very desirable advance in this field.

**Specific comments**

1. Negative results w.r.t. aerosol size and type should also be mentioned in the abstract.

2. P. 2, line 14: please provide at least one general reference about D.A. for atmospheric models.

3. Section 3.2: please clarify the repartition of the analysis increment into aerosol types. The end of the section states "...*we decided to keep constant the relative mass contributions. After the analysis increment is calculated, it is repartitioned to the different bins in the model according to their background fractions of the total aerosol mass.*" Does this apply to aerosol types as well? This could be a key information w.r.t. the negative results which are suggested for addition in the discussion.

4. Equation (7): AOD is obtained by summation over all size bins and levels – but where is the summation over aerosol types? Shouldn't extinction cross-sections be explicitly noted as

depending on aerosol type as well?

5. P. 8, lines 8-10: specifying correlation lengths in terms of geographic degrees leads to assimilation increments which become increasingly smaller in longitude as the assimilated observation gets closer to the poles. This is not a concern for the present study which is limited to the Mediterranean domain, but may be worth mentioning nonetheless.

6. Section 6.2: this could be the best place to document the negative result w.r.t. aerosol type, e.g. with a timeseries figure similar to figure 3, but showing the contribution of each aerosol type to AOD in the assimilation experiment. If I understand well, all aerosol types increased during the desert dust episode – contrarily to expectations.

7. Figures 1 and 9: a novice reader could confuse the "forecast" results with those of the direct model run. Consider using instead the words "one-hour forecast" or even "first guess".

8. P. 14, lines 8-9: *"The stations in the east, like in Lampedusa and Cyprus, were not influenced by these dust events. They are mostly influenced by sea salt aerosols, and the data assimilation also here has a very positive impact."*
   This could suggest that DA has a very positive impact on the amount of sea salt aerosols at these stations. I understand that this is not the case, so a more precise formulation is necessary, e.g.: *They are mostly influenced by sea salt aerosols, and the data assimilation has a very positive impact on AOD at these stations as well.*

9. P. 21: the discussion does not describe Fig. 10 clearly. For example, you write "*...the assimilation on 26 and 27 June lowers the intensity in the higher layers...*" but from pane c) it appears that assimilation lowers intensity in **all** layers on these days!?
   *"On 30 June, the assimilation increases even more the relative difference between the layers. As a result, aerosol mixing ratios are larger in lower layers than in higher layers, while in the direct model run this is the opposite."* I am unable to see this.
   At first sight this figure is contradictory with the AOD comparisons which showed elevated aerosol amounts in the analyses and observations. Specifically, the maps on 29 June (Fig. 2) showed that at the corresponding trajectory location (Fig. 10d: off the coast of Morocco) the AOD is larger in the analyses than in the direct run. Hence you should explain why the mass mixing ratio becomes smaller nonetheless.
   Overall, Fig. 10 is pretty but it shows only model results while the journal is AMT (not GMD). Since the result seems hard to discuss, you could as well drop it. This way the paper would finish with Fig. 9, which displays the most important outcome.

10. P. 23, line 2-3: *"To take into account these uncertainties in the assimilation process, we defined the variances in the matrix B in such a way that it allows a margin for the model error (Talagrand, 2003)."* This is not clear. Does it refer to setting the model errors twice as large as the observations (24% versus 12%) as explained p.8, lines 1-6? If yes, please re-write more explicitly. If not, please expand section 3.4 accordingly.

11. P. 23, lines 6-7: *"One would be to add an additional term in the cost function where we would describe the errors in the model evolution."* Please provide a reference on this

technique. I believe that this it is named "Variational Bias Correction" at ECMWF.

12. P. 23, lines 12-18: One wonders why you give these details about error characterization in Benedetti et al. (2009). Do you mean that these implementation details differ from yours, yet did not prevent that *"Their 4D-Var analysis results showed qualitatively a very similar impact of assimilation as in this study."*? If yes, please state more clearly what are the differences between your implementation and theirs. You could also propose a few ideas about the added value of your approach compared with the implementation at ECMWF, e.g. computing cost of 3D-FGAT lower than 4D-Var ? Or shorter assimilation cycles (here 1 hour versus probably 12 hours at ECMWF) which could be necessary to improve vertical distribution as shown on Fig. 9c ?

13. P. 23, lines 19-28: here you discuss the impact of assimilation on size distribution, but you chose to remove any such comparison from section 6. Hence this whole paragraph is not supported by any figure and actually becomes quite unclear. This is similar to the issue of aerosol type speciation which I raise in the general comments, but less important in my opinion. So you could either re-insert the size distribution results in section 6 and discuss them there, or drop this paragraph altogether.

14. P. 24, lines 25-28: you propose simultaneous assimilation of AOD by different satellite instruments, but in this case the inter-instrument biases need to be carefully considered and corrected first, before any assimilation.

15. P. 25, line 1: *"Then with this information we could modify the size distribution and aerosol bin distribution in the model"*. What is the difference between size distribution and aerosol bin distribution? Couldn't this information simply allow you to modify the partitioning between aerosol types in the model?

**Minor comments**

- Citation style is sometimes erroneous, with in-text citation (LaTeX command `\citet`) used insetad of in-parentheses citation (LaTeX command `\citep`). Exemples: p. 2, lines 18-19; p. 5, line 13; p. 7, line 15.

- P. 7, line 23: The $\chi^2$ test does not "define" errors. Consider: "The $\chi^2$ test is an a posteriori diagnostic which allows to check that the errors are properly specified. It checks if, for each assimilation window..."

- P. 8, line 4: *"Therefore, the possibly smaller AOD..."*

- P. 10, line 20: *"The assimilated model can more readily lower the overestimated values than  elevate the underestimated values."*

- P. 18, last line: *"To further explore this, we compare the modelled and the measured vertical profile follows"*. Please re-write the sentence.

- P. 20, line 5: *"LOAC measurements acquired during the **balloon** flight* (**Fig. 9b**) *are colocated with..."*

---

## Author Comment (AC1) · 20 Sep 2016

**Response to the comments of the Anonymous Referee #2**

**Sič et al.**

September 20, 2016

We thank the reviewer for detailed and thoughtful comments that make the article clearer and more complete. The reviewer suggested some interesting complimentary ideas to include in the paper. Some of them we did (trajectory analyses of selected air parcels), and some of them we already considered to be part of the companion paper that will follow this one. The second paper will concern with the lidar profile assimilation in our model.

Below we indicate for each individual comment how we have dealt with it.

Specific comments:

1. "reduced the bias in the AOD (from 0.050 to 0.006) and increased the correlation (from 0.74 to 0.88)": please state for which dataset (MODIS, SEVIRI, AERONET). Also, include a standard deviation or RMS error. If any spatiotemporal averaging was done beforehand, specify so now.

We changed the abstract as suggested by the reviewer.

2. Neither Textor nor Lee consider efforts to improve models. Textor evaluates an ensemble of models, while Lee developed a methodology to better understand a single model. True, all of this with a goal to finally improving the models but neither of these papers actually gets there. There is a vast literature out there about model evaluation and development, and more (and more relevant) references should be included.

When referencing this sentence, we did not have in mind just to mention the studies that directly focus and improve certain aspects of some particular models, but referenced more general papers that illustrate the diversity of efforts to improve aerosol models. Nevertheless, we understood the remark of the reviewer and added some additional references that cover this topic from different aspects (for example Kanakidou et al., 2005; Fuzzi et al., 2006; Vignati et al., 2010; Boucher et al., 2013).

```
3. "diversity of parameterizations": there are many
causes (e.g. dynamical cores, emissions, parameter
uncertainty) for model diversity, parametrisation of
processes is only one of them.
```

We corrected the text as suggested.

4. Please include at least the following: Collins et al JGR 2001 (OI), Weaver et al JAS 2005, Generoso et al JGR 2007 (3D-VAR), Zangh et al JGR 2008 (3DVAR), Yumimoto et al. ACP 2008 (4D-VAR), Sekiyama et al. ACP 2010 (EnKF), Schwartz et al. 2012 JGR (3D-VAR), Saide et al ACP 2013 (3D-VAR), Rubin et al ACP 2016 (EnKF)

We included the majority of these references dealing with the aerosol assimilation. The idea was to mention efforts of different groups in different models, for that we included the names of the models the groups worked with.

5. "The particle size distribution for each type is divided into 6 size bins, characterized by the particle average diameter and mass. Each aerosol bin is then treated as a passive tracer: aerosols are emitted, transported and removed from the atmosphere, however there are no transformations or chemical reactions between aerosol types, between size bins or with gases." To put it differently: a 2-moment scheme (mass & numbers) is used to describe aerosols in each bin but no coagulation or condensation processes are modeled? If these latter processes are not considered, wouldn't a simple mass scheme do just as well?

The work on secondary aerosols was done in parallel with the developments on the assimilation module. It means that we need a two moment scheme in the model and in the assimilation runs since the coagulation and the condensation processes will be included. As we talk about the performance with a bulk mass scheme, it would be needed to evaluate its performance, but that is not one of our priorities. The effect of different size-depended processes are taken into account explicitly for each bin in our aerosol scheme, and, for sure, that is advantageous compared against a bulk mass scheme.

6. "We use daily BB emissions for better synoptic forecasts, which is not possible with the monthly mean emissions of Lamarque et al. (2010)." But this begs the question how a diurnal cycle in emissions would affect your standard model run (and your DA results for that matter). Emission datasets with (imposed) diurnal cycles for anthropogenic emissions exist for the European domain, e.g. TNO-INERIS. Why were such datasets not used? The need to use a database that estimates emissions of particular events was identified, since we knew that in the region of our analysis there had been active forest fires. The daily GFAS data provided us an accurate estimation of the emitted carbon aerosols in the region during the experiment period. In the end, it turned out that these aerosols had a limited impact in terms of the AOD in the regions (dominated by desert dust aerosols), and that was also true along the path of the measurement flights.

"The model is then run over a cycle length (1 hour) to obtain the analysed trajectory." I could not find any clear statement over which domain observations are assimilated. Sect. 4 seems to suggest only over MEDIO2. But why then a cycle of 1 hour? New MODIS observations will happen only twice every day (at best). Since the cycle is only 1hr, isn't 3D-FGAT very similar to standard 3D-VAR?

We performed a 3D-FGAT assimilation with cycle of 1 hour over the MEDI02 domain. A shorter window in 3D-FGAT is advantageous since in this method the evolution of the model is not taken into account. But, due to the characteristics of our assimilation system, another reason to use such a short cycle is the fact that in this case we do not average the mass fractions of the bins (which would be the case in longer assimilation windows), which is certainly advantageous.

Also, we changed the text to make clear that MEDI02 is the domain in which we assimilate data (P3 L30, P10 L27).

P5, L13: Reformat "Courtier et al. (1994)"

Corrected.

P6, L3: "nor does it need to contain the inter-bin covariances"

Corrected as suggested.

P6, L6: "could be weighted by different quantities, like number or mass concentration, or extinction coefficient". I disagree. If the innovation is zero (i.e. forecast AOT agrees with observations), it makes sense that the relative distribution of aerosol species does not change. This limits how you can redistribute aerosol. I also don't understand how number and masses would make a difference in your system. From an earlier description, I gather that masses and numbers in each bin are closely related (since there are no inter-bin processes or gaseous condensations). As the innovation is in terms of the total aerosol concentration, if it is non zero, it is necessary to relatively fix it related to some model quantity. number and masses are close related as there are no secondary aerosols and inter-bin processes, but their relative ratios are not the same, and reflect the differences of the number and mass size distribution of the same population of aerosol particles.

P6, L20: "also by taking into account the hygroscopicity of sea salt aerosols" Surely the hygroscopicity of sulfate and organic carbons is considered as well? This is part of the OPAC (by the way, I believe the authors may be mistaken the climatology GADS for the aerosol optical properties database OPAC)

As only having the primary aerosols in the model runs, the primary organic carbon is considered as non hydrophilic. Strictly speaking, the organic carbon particles are both primary and secondary aerosols, but in this paper we only have primary carbon organic aerosols. As for the organic carbon, we refer to the carbon content in the organic material, as it is often the case (for example, Seinfeld and Pandis, 1998). GADS (Global Aerosol Data Set) and OPAC (Optical Properties of Aerosols and Clouds) are closely related. To acknowledge this, we have included a reference for OPAC as well in the new version of the paper (P7 L4).

P6, L27 and further: I am not sure what this adds to the paper. Consider removing it or clarifying its importance.

Development of the observation operator, and its tangent-linear and adjoint version, takes a big part in the development of the assimilation system. Their characteristics and correct functioning can be induced by running the tangent-linear and adjoint tests, which are briefly presented in these lines. This part is often neglected in similar papers as it is not directly connected with the main study conclusions, but because it is an important part of our development we decided to mention it. However, we clarified the text of the new version of the paper (P7 L11 L15).

P7, L16: "defines the observation and representativeness errors. These errors are considered to be non-correlated, which means that all non-diagonal members (covariances) in the matrix R are zero". This is an assumption that is unlikely to be true, especially given that your super-observations are aggregates over 0.2 by 0.2 degrees (a very short distance). At the very least you need to state this, better yet if you can provide justification for your assumption. The correlation of the observation errors most probably exist to some degree, but we consider errors to be non-correlated (neglect them) due to a lack of correlation statistics and difficulties in implementation, which is a strategy pursued by many studies. Super-observations might have a certain effect in reducing the correlation of the observation errors, but that is not our main supporting point for neglecting them. We clarify the text in the updated version of the paper (P8 L3).

P8, L1: "As the optimal parameters, we estimated that the percentage for the errors of the model should be twice as large as for the observations (24% and 12% respectively)." I don't understand what the 24 and 12% refer to. R should be based on realistic assessment of the observational error (including representativeness error), independent of the data assimilation. So is 12% a typical observational error? But it seems estimated from Eq 13, so it is not independent from the data assimilation system.

The text dealing with B and R error variance was not clear enough and it was misleading concerning the definition of the R matrix error variance. The observation errors are defined as a percentage of each available observation, and the percentage that we use (12%) is estimated from the known MODIS uncertanties (Remer et al, 2005). The B matrix error variance is estimated with the help of the equation 13. We made necessary changed in the text to clarify all this (P8 L7 - L12).

P8, L10: "The implemented horizontal correlation length is 0.4." This is based on other studies or just an educated guess?

We estimated this value by taking into account the model resolution, the density of assimilated observations, and the experience from previous runs we have done in the system (for example, Barré et al, ACP, 2012). We want to limit it to not influence negatively the minimisation process because of overlapping influence of the observations (and to not make possible artefacts in the analysis due to too dense observations), but having the value which will be realistic and which will in the same time help the minimisation process.

P8, Section 4: It appears that outside MEDIO2 no observations are assimilated? Why not? Wouldn't you want to constrain inflows across the MEDIO2 boundaries?

We assimilate observations only in the MEDI02 domain. The multi-domain assimilation in our system is not yet implemented. By assimilating the observations in the regional domain, we have the model resolution in the domain higher than in the global domain  $(0.2^{\circ} \text{ vs. } 2^{\circ})$ . The domain covers a geographic region with borders that are quite distant from our region of interest (the Mediterranean basin and North Africa). Thus, this way we limited the influence of the model inflow, from the global domain in which we do not assimilate the aerosols, to our region of interest.

P8, L20: MODIS C5 is known to have systematic biases, e.g. due to windspeed and cloud cover over ocean but also over land. This has greatly improved in C6 so why did you not use this? Also, these error estimates are very optimistic (see Fig 20 in Schutgens et al AMT 2013). Even C6 has higher error estimates.

Earlier, in the previous studies in our system, we remarked the biases in the MODIS C5, especially related with a cloud cover in cloudy regions, like at midlatitudes over oceans (for example, Sic et al, 2015). But we did not notice any problems with Collection 5 over our region of interest. In the transition period, we opted for the C5. Regarding our choice of the errors, for the minimisation process, the ratio between the background and observation error is more important than the values themselves. And as already described, we made different tests to estimate optimal ratio between the two.

P8, L25: Earlier the authors mention representativeness errors. Do you assume these are essentially zero because of the creation of super-observations? But you will seldom have grid-filling MODIS observations so representativeness may still be an issue. Also, over distances of less than 50km aerosol is usually strongly correlated. As a result, retrieval errors and representativeness errors may also be strongly correlated. Yet you assume they are not. Please discuss this.

As already mentioned before, we presume that super-observations might have a positive effect on reducing the observation error-correlations, but that is not our argument to neglect the correlations. It is rather the fact that we do not know the error statistics, which is necessary to define them. Assuming that errors are non-correlated is an usual strategy. Super-observations certainly reduce random and representativeness errors to some degree, and this effect depends on the number of the observations averaged in each surface gridcell (in our case about max 4-5 observations per cell). But we presume that representativeness errors are not of a big concern in our case, since the resolution of the instrument (about 10km in nadir) and the resolution of the model  $(0.2^{\circ})$  are not that different. Also, swath observations do not overlap between themselves, and by averaging them we should have some sort of a complementary effect. On the other hand, assimilating all available L2 observations in the system would have rather negative effects for the minimisation and the analysis. The possible choices we have is to make super-observations or to do a data thinning (which does not impact random and representativeness errors). Although not crucial, both methods should have some positive effect in reducing the correlation of correlated errors (for example, Liu and Rabier, 2003, QJRMS and references therein).

P9, L5: What are typical SEVIRI AOD errors over ocean? Did you consider assimilating these data (which have a much higher time resolution than MODIS)?

Seviri data has a higher time resolution, but also a higher error. Concerning the error of SEVIRI prodict, for exemple, Breon et al (2011) used as a statistical indicator, the fraction of retrieved AOD that falls into MODIS' defined accuracy (over ocean  $0.03 + 0.05\tau$ ) when compared to the sunphotometer data. The computed fraction for MODIS over ocean was 57%, while 40% for Thieuleux et al. SEVIRI product. We made some tests by assimilating the SEVIRI data in the system. In the end, for our case, the MODIS data with two passes over each point per daytime provided enough data to the assimilation system to have a good coverage of the whole region, and higher time resolution would not have a strong impact on our analysis. Also, it does not exist a common SEVIRI product covering both ocean and land. The solution is to combine two products, which we tested, but then we encountered a strong discrepancy between the two products at coastlines.

P9, L8: 3m and 5m should likely be 3km and 5km

Correct. Corrected in the text.

P9, L12: Why did the authors use AERONET L1.5 data? Since their experiment is from 2012, L2 data are available and generally far superior in cloud screening. It seems they do not consider AE as a observation for evaluation. Why not? Have they considered using AERONET inversion data for evaluation? SSA measurements would allow for an interesting evaluation of aerosol speciation.

The version of AERONET data that we used changed from the very first draft of the paper. We passed from L1.5 to L2, and indeed, all the presented aeronet results we made using the L2 data. Now, the description of the data in the text also mentions the correct version of AERONET data. In this study we focused on the total aerosol concentration, which was our control variable, and not so much on the size distribution and aerosol specialisation, because we have fixed the size distribution in each assimilation cycle by using the constant mass fraction between the bins and not have the complete family of aerosols implemented in the model where we assimilated. But certainly, looking at the data that gives information about the size or the type of aerosol could give more valuable insights of the model and assimilation reaches, and we would like to deal more with theses aspects in the follow-up paper.

P9 & 10 PCASP and LOAC: is anything known of measurement errors for these instruments (after the time-averaging)?

Cai et al. (AMT, 2013) reported the Poisson error for PCASP as 5 - 15%, and Renard et al. (AMT, 2015) reported the LOAC total concentration measurements as  $\pm 20\%$  when concentrations are higher than  $1 \, cm^{-3}$  and up to about  $\pm 60\%$ when concentrations are smaller than  $10^{-2} \, cm^{-3}$ . We add this information in the new version of the paper (P10 L14 L20).

P10, L12: Is there a separate spin-up of the assimilation as well? Or do the authors start evaluating results after the very first assimilation cycle?

The last 10 days of the spin-up are the so-called assimilation spin-up for the assimilation model run. We added this information in the text.

P10, L13: "The assimilation cycles in the experiment have a length of one hour." I guess most observations are made near the same time so why use a one hour cycle?

We responded to this comment also in the response 7.

P10, L12: "(directly forecasted" Is this the direct experiment or the forecast that is part of the assimilation experiment? P10, L22: This suggests that biases are small (RMS error standard deviation) and that suggests you are comparing forecast and analysis after some spin-up of the assimilation system? We compared the assimilation forecast and the assimilation analysis fields of each assimilation cycle with the observations that we have assimilated. There is a spin-up of the assimilation system also, as mentioned in the response 22. We clarified this point in the new version of the paper (P10 L30). Similar values of the RMSE and standard deviation indeed suggest that biases are not important.

P12, L3: "MODIS overpasses each point twice" Undoubtably the authors are aware there are two MODIS instruments, each passes over a point once a day.

We clarified this sentence in the new version of the manuscript (P13 L15).

P 13, Fig 4: How do the authors interpret the difference in direct model run for MODIS (Fig 1) and SEVIRI (Fig 4). The first one seems unbiased but in the second one the model is clearly biased low. Is this because Fig 1 really compares the forecast and not the direct model run? If so, how does the direct model compare against MODIS and does it look similar as Fig 4? This interpretation is correct. The direct model run underestimated the intensity of two desert dust outbreaks that dominated the evaluation period. This underestimation can be seen in the comparisons with different datasets, SEVIRI in Fig. 3 abd Fig. 4, AERONET in Fig. 6 and Fig. 7, and MODIS in Fig. 1.

P 18 Fig 8: I find this very interesting and a major result of this paper. Maybe panel d) can be changed to include assimilated observations during the day of the flight? It would help understand what information was available and what not.

We changed the panel d) by adding the assimilated observations during the day of the flight. This information can help to understand what was the most recent impact of the observations in the model, but one should be careful with it, since it does not tell us about the influence of the observations of the previous days prior to the flight.

P20, L3: "Sometimes, the generalized multiplicative change of the aerosol profile in the assimilation can produce unsatisfactory effects in some layers. For the second flight, although the concentrations in the plume are hugely improved, near the surface the increase of aerosols lead to a larger overestimation in the model." To me it seems these two sentences make more sense if their order is reversed.

We reversed their order in the text.

P20, L14: "This demonstrates that the continuous assimilation of good quality AOD observations can correct a shape of the aerosol vertical profile, although a single AOD assimilation cycle can only expand or shrink the profile shape (as the AOD observations do not contain the information on the vertical). For the profile in Fig. 9c, by comparing the forecast and the analysis of the same assimilation window, we see that the single AOD assimilation cycle expands the profile but does not change its shape, what multiple cycles does." I agree with this assessment and think it shows nicely the power of data assimilation. However, the authors may want to reconsider earlier statements like P20, L6 "the aerosol concentration in the assimilation run at a certain height can be correct only if the profile shape is well simulated in the direct model run." which is far more restrictive.

We agree with the reviewer. There were also a few more restrictive sentences like this in the text. We have modified them to limit this conclusion to only one assimilation cycle (P18 L11, P21 L3).

P20, L18: "multiple cycles" it may be useful to point out to readers that it is not just the multiple assimilation cycles that improve the profile but also assimilation throughout a spatial domain. So dust originating from different regions and transported at different levels is adjusted in previous cycles before it ends up in the main plume. One question is then how important the DeepBlue observations are to your assimilation. I expect quite important because they contain information closest to individual dust source regions.

We agree with the reviewer, and it is really the model propagation of increment that can improve the aerosol profile during the AOD assimilation. It is necessary that the increment comes from different directions (or even the same direction but only covers different distance). We have changed couple of sentences making this mechanism clear. The multiple/multiday assimilation is also important as it can amplify the effect of the profile change by subsequent cycles as we showed in Fig. 10, and of course, it makes situation of the profile change more plausible to occur, as we assimilate for the prolonged period of time. Related to the importance of the DeepBlue observations, in our case the desert dust plumes took a quite unusual trajectory (as shown in Fig. 10d) which made their covered distance and time to the Mediterranean bigger. In such case the multiple passes of the MODIS instruments would be probably sufficient for the assimilation to strongly influence the plume. The plume was 3-4 days above regions where the ocean and the land MODIS products were available. We assume that, in such case, the DeepBlue product is not that important, contrary to cases where the source of the desert dust emission is much closer to the Mediterranean sea and where the distance and time to get to the basin is smaller. In such situations, the assimilation of the DeepBlue product would have a crucial impact on the assimilated field.

P20, L23: "Therefore, we cannot follow the dust plume by following an air parcel." I think I understand this sentence but the dust is still advected by the winds: following air parcels at different heights should give you the dust plume or all Lagrangian analyses are useless (HYSPLIT or FLEXPART).

We cannot say that the Lagrangian analyses for aerosols are useless; but one should have in mind that following an air parcel and an aerosol particle is not the same. On the aerosol particle act physical processes that do not act on the air parcel. Of course, aerosol particles are advected by the winds as well, but the Lagrangian trajectory can tell us only about the direction from/to where an aerosol particle is coming/going but cannot tell us more than that. There is no accurate information about a region where it was/will be in some time back/forward. This is even true for an air parcel itself, but it even lot more for aerosol particles. In Fig. 10d, to obtain the trajectory of an aerosol particle as it is modelled/presented would not be possible just by following the air parcel. It is necessary to follow the particle itself. After a couple of days, the trajectories of an aerosol particle and an air parcel would differ. All due to the processes acting differently on the particle, and on the air parcel.

P21, Fig 10 (panel d): is it possible to somehow plot Lagrangian trajectories as well to show how different air parcels move before contributing to the plume? This would be a very strong visual support for the nice analysis in 6.6

We added subfigure d) to Fig. 9. It shows the different directions of the origins for the parcels at the different heights. Especially, what was important in our case, the different directions between the layers nearer to the surface, and the layers where were present desert dust particles.

P22, L9: "In doing so, our assimilation system showed to be more efficient in lowering overestimated AOD values in the model than increasing underestimated values" This has been shown in several other assimilation studies as well. It's due to model errors being defined relative to the model forecast which is unavoidable if you assume the model to be unbiased. I expect your solution (in L14) would not work: if the model substantially underestimates actual AOD (i.e. outside the model's error envelope), adjusting observational error will not help (not to mention that it is close to fudging).

We agree with the reviewer about the reason of this effect, as we had also presented it in the text. We agree, as well, with his/her remark for the case when the model strongly underestimates the AOD, the mild adjusting (lowering) of the matrix R cannot fix the problem efficiently. This approach would lower the weight of the observations for such point, but the background error would stay significantly lower that the observation error, and the change of the weight between background and observation in the cost function would not greatly change. We modified in the text this sentence to make clear that the effect of this approach is be limited.

P23, L12: Also, Benedetti uses a 4D-VAR method not 3D-FGAT

We already mentioned it in the last sentence of the paragraph. We added it again, but earlier in the paragraph (P24 L15).

23, L19-28: This is the first time that mention is made of the very different size ranges of model and observations. The authors should state the model range (I could not find it in their paper) and how they dealt with the discrepancy (I assume they only compared number concentrations in overlapping bins). Ideally, this should be described in resp. the model and observation section but not here in the discussion. Instead of this long paragraph a single sentence that there was insufficient information to compare size distributions should be sufficient.

We added a table with the description of all size bins in MOCAGE in the section about the model description in the new version of the manuscript (Table 1). Further, we compare the model and the observations only in overlapping bins, and the considered ranges in our result analysis were mentioned in the figure descriptions for both the PCASP (Fig. 8) and the LOAC instrument (Fig. 9). We dropped this paragraph out of the text.

P24,L6: Instead of 'influence' I suggest 'improve'

We changed the text following the suggestion of the reviewer.

---

## Author Comment (AC2)

**Response to the comments of the Anonymous Referee #3**

Sič et al.

September 20, 2016

We thank the reviewer for detailed and thoughtful comments that make the article clearer and more complete. Following the suggestions, we changed and added some information, and put more emphasis and explanations where it was necessary.

First of all, we would like to explain the "lack" of aerosol speciation influence by the assimilation in this study. First, the reviewer called it as a negative result, but later in Comment 3 the reviewer correctly interpreted what could be the reason of such system behaviour. The repartition of the increment that is explained in the Section 3.2 applies to all bins and species (in total) that are present in the study. Although, for example, the desert dust was the dominant aerosol type during the studied dust episode, the increment is repartitioned according to relative mass fractions of all bins present in the model (4 species with 6 bins each). Thus, the speciation of aerosol species and sizes that is present in the model field before an assimilation cycle remains the same also after the cycle and the assimilation does not change relative ratio between species and their sizes.

To change the ratio between species before and after an assimilation cycle, it would be necessary to have as a different control variable, like the 3D concentration of all 24 bins (4 species with 6 bins), and not the total concentration (all bins summed) as we have. In that case the system with its tangent linear, and adjoint code would automatically decide about the increment of each species (bin to be more precise) separately. The problem with this approach is that if we assimilate aerosol optical depth, in observations there is no information about separate species and the system would decide about repartition between different species without real foundations based on observations. To have this approach fulfil its potential it would be necessary to assimilate the speciated aerosol observations.

In the text we wrote that "AOD assimilation can also impact aerosol size and type for the same reason, but this was not evident in this experiment", thinking only about the effect of the multicycle assimilation and the model propagation of the increment. The eventual change of aerosol size and type depends on the model propagation by winds and in the same time on the differential effects of the physical processes on aerosol size and type; and it is expected to be smaller and more difficult to detect than the profile improvement which depends only on the model propagation of the increment. Its detection would necessitate more type and size dependent measurements, and the secondary aerosols in the model, and therefore it was out of the scope of this study.

We modified the text with the increment repartition to make it clear.

Concerning specific comments, below we reply point to point to the reviewer comments.

> 1.  Negative results w.r.t.  aerosol size and type should
> also be mentioned in the abstract.

As the reviewer correctly interpreted in Comment 3, it is the increment repartition in the system that fix the relative proportion between aerosols inthe model, including size and type.

> 2.  P. 2, line 14:  please provide at least one general
> reference about D.A. for atmospheric models.

We added two of pioneering references related to the assimilation of the atmospheric constituents, Fisher and Lary (QJRMS, 1995) and Elbern et al. (JGR, 1997)

> 3.  Section 3.2:  please clarify the repartition of the
> analysis increment into aerosol types.  The end of the
> section states "...we decided to keep constant the relative
> mass contributions.  After the analysis increment is
> calculated, it is repartitioned to the different bins in
> the model according to their background fractions of the
> total aerosol mass." Does this apply to aerosol types as
> well?  This could be a key information w.r.t.  the negative
> results which are suggested for addition in the discussion.

We agree with the reviewer, this is a key information. Since we keep constant relative amount of all bins (types included) we limited the change of the size or type by assimilation, as explained above in more details.

> 4.  Equation (7):  AOD is obtained by summation over all
> size bins and levels { but where is the summation over
> aerosol types?  Shouldn't extinction crosssections be
> explicitly noted as depending on aerosol type as well?

We mean by "all bins", all different bins of all available aerosol types in the model. And, therefore, extinction cross-sections directly depend on aerosol type as well. We clarified this in the revised version of the manuscript (P6 L24 L30).

> 5. P. 8, lines 8-10: specifying correlation lengths in
> terms of geographic degrees leads to assimilation increments
> which become increasingly smaller in longitude as the
> assimilated observation gets closer to the poles. This
> is not a concern for the present study which is limited
> to the Mediterranean domain, but may be worth mentioning
> nonetheless.

The horizontal correlation lengths in the model are constant over the whole domain. The value in degrees is converted in kilometres (at equator), and then applied to the whole domain. Therefore, in the revised text, instead of presenting it in degrees we have changed it to kilometres (P8 L25).

> 6. Section 6.2: this could be the best place to document
> the negative result w.r.t. aerosol type, e.g. with a
> timeseries figure similar to figure 3, but showing the
> contribution of each aerosol type to AOD in the assimilation
> experiment. If I understand well, all aerosol types
> increased during the desert dust episode { contrarily to
> expectations.

As explained in the general comments section, the result is the direct outcome of the choice to assimilate total concentrations and repartition the increment to all bins (types included). Therefore, during the desert dust episode, by assimilation we increase/decrease all aerosols present in a grid boxes.

> 7 Figures 1 and 9: a novice reader could confuse the
> "forecast" results with those of the direct model run.
> Consider using instead the words "one-hour forecast" or
> even "first guess".

We agree with the reviewer, and we modified the text as suggested (Figs. 1 and 9).

> 8 P. 14, lines 8-9: "The stations in the east, like in
> Lampedusa and Cyprus, were not influenced by these dust
> events. They are mostly influenced by sea salt aerosols,
> and the data assimilation also here has a very positive
> impact." This could suggest that DA has a very positive
> impact on the amount of sea salt aerosols at these stations.
> I understand that this is not the case, so a more precise
> formulation is necessary, e.g.: They are mostly influenced
> by sea salt aerosols, and the data assimilation has a very
> positive impact on AOD at these stations as well.

The assimilation can increase/decrease the amount of aerosols already present in a grid box. In the case of Lampedusa and Cyprus stations, since we assimilate

all aerosol types and since sea salt aerosols were dominant before the assimilation, it is mostly this type of aerosols that the model corrected.

> 9 P. 21: the discussion does not describe Fig. 10
> clearly. For example, you write "...the assimilation on 26
> and 27 June lowers the intensity in the higher layers..."
> but from pane c) it appears that assimilation lowers
> intensity in all layers on these days!? "On 30 June, the
> assimilation increases even more the relative difference
> between the layers. As a result, aerosol mixing ratios are
> larger in lower layers than in higher layers, while in the
> direct model run this is the opposite." I am unable to see
> this. At first sight this figure is contradictory with the
> AOD comparisons which showed elevated aerosol amounts in the
> analyses and observations. Specifically, the maps on 29
> June (Fig. 2) showed that at the corresponding trajectory
> location (Fig. 10d: off the coast of Morocco) the AOD is
> larger in the analyses than in the direct run. Hence you
> should explain why the mass mixing ratio becomes smaller
> nonetheless. Overall, Fig. 10 is pretty but it shows only
> model results while the journal is AMT (not GMD). Since the
> result seems hard to discuss, you could as well drop it.
> This way the paper would finish with Fig. 9, which displays
> the most important outcome.

Since in Fig. 10 we can evidently see the profile change on the larger part of the plume trajectory, we decided to keep it, but we modified the text to clarify the message (P21 L28 - P22 L5). Figure 10 shows only the model runs, but it should be seen rather as the follow-up of Fig. 9. We explain the mechanisms on how assimilation can change the profiles, and one can see an obvious change of the aerosol profile and its evolution during the lifetime of the plume, from its emission to its dissipation.

We would like also to explain that Fig. 10 and Fig. 2 are in good agreement. On 29 June, in the analysis, the mass mixing ratios between the higher and lower layers are comparable, but the lower layers contain a lot more aerosol mass since the mass mixing ratio is an air density dependent unit. Thus, lower layers give majority of AOD in this case, and in the direct model run we see that mixing ratios are smaller near the surface than in the assimilation run.

We presented the figure in mass mixing ratio as this unit lowers the difference between higher and lower levels (because it is pressure dependent) and we can easily follow the evolution of the aerosols in the plume at higher levels and aerosols nearer to the surface in the same figure.

> 10 P. 23, line 2-3: "To take into account these
> uncertainties in the assimilation process, we defined the
> variances in the matrix B in such a way that it allows a
> margin for the model error (Talagrand, 2003)." This is not
> clear. Does it refer to setting the model errors twice as
> large as the observations (24% versus 12%) as explained p.8,
> lines 16? If yes, please rewrite more explicitly. If not,
> please expand section 3.4 accordingly

Yes, it refers to the percentages we used to define the variances. The text is corrected as suggested (P24 L3).

> 11 P. 23, lines 6-7: "One would be to add an additional
> term in the cost function where we would describe the errors
> in the model evolution." Please provide a reference on this
> technique. I believe that this it is named "Variational
> Bias Correction" at ECMWF.

We mentioned also the bias correction method at P24 L10, but what we had in mind is the method referred as the weak constrain 4D-VAR. There is really another term in the cost function $J_q$ that accounts for the model errors with the corresponding model error covariance matrix Q besides the background term $J_b$ and the observation term $J_o$. The bias correction method does not change the cost function, but the known biases in the model attributes to the observation error in the system who in such way account them as well.

> 12 P. 23, lines 12-18: One wonders why you give these
> details about error characterization in Benedetti et al.
> (2009). Do you mean that these implementation details
> differ from yours, yet did not prevent that "Their 4D-Var
> analysis results showed qualitatively a very similar
> impact of assimilation as in this study."? If yes, please
> state more clearly what are the differences between your
> implementation and theirs. You could also propose a few
> ideas about the added value of your approach compared with
> the implementation at ECMWF, e.g. computing cost of 3D-FGAT
> lower than 4D-Var ? Or shorter assimilation cycles (here
> 1 hour versus probably 12 hours at ECMWF) which could be
> necessary to improve vertical distribution as shown on Fig.
> 9c ?

The motivation to explicitly give details of the Benedetti et al. (2009) study is that our studies are similar as we both used total concentrations as the control variable, and the similar results show that the differences in our system do not influence qualitatively the analysis. Of course, we did not intend to go the detailed comparison, but to point out the similar performances in our system that are influenced by the choice of the control variable.

> 13 P. 23, lines 19-28:  here you discuss the impact of
> assimilation on size distribution, but you chose to remove
> any such comparison from section 6.  Hence this whole
> paragraph is not supported by any figure and actually
> becomes quite unclear.  This is similar to the issue
> of aerosol type speciation which I raise in the general
> comments, but less important in my opinion.  So you could
> either re-insert the size distribution results in section 6
> and discuss them there, or drop this paragraph altogether.

Considering also a similar comment from the Reviewer 2, we have dropped this paragraph out of the manuscript.

> 14 P. 24, lines 25-28:  you propose simultaneous
> assimilation of AOD by different satellite instruments,
> but in this case the inter-instrument biases need to
> be carefully considered and corrected first, before any
> assimilation.

We agree with this remark and we added it to the revised text (P25 L20).

> 15 P. 25, line 1:  "Then with this information we could
> modify the size distribution and aerosol bin distribution
> in the model".  What is the difference between size
> distribution and aerosol bin distribution?  Couldn't this
> information simply allow you to modify the partitioning
> between aerosol types in the model?

We simplified the text, and we mention only the term 'size distribution'. This information from the multi-wavelength observations in certain cases could be used to partition between aerosol types as well.

> Minor comments...
>     Citation style is sometimes erroneous, with intext
> citation (LaTeX command citet) used instead of inparenthesis
> citation (LaTeX command citep).  Examples:  p.  2, lines
> 18-19; p.  5, line 13; p.  7, line 15.
>     P. 7, line 23:  The $\chi^2$ test does not "define" errors.
> Consider:  "The $\chi^2$ test is an a posteriori diagnostic which
> allows to check that the errors are properly specified.  It
> checks if, for each assimilation window..."
>     P. 8, line 4:  "Therefore, the possibly smaller AOD..."
>     P. 10, line 20:  "The assimilated model can more
> readily lower the overestimated values than to elevate the
> underestimated values."
>     P. 18, last line:  "To further explore this, we compare
> the modelled and the measured vertical profile follows".
> Please rewrite the sentence.
>     P. 20, line 5:  "LOAC measurements acquired during the
> balloon flight (Fig.  9b) are colocated with..."

We have corrected all pointed errors.